# Guideline adherence in speech and language therapy in stroke aftercare. A health insurance claims data analysis

Daniel Schindel[1]*, Lena Mandl[1], Ralph Schilling[2,3], Andreas Meisel[4], Liane Schenk[1]

1 Institute of Medical Sociology and Rehabilitation Science, Charité – Universitätsmedizin Berlin, Freie Universität Berlin and Humboldt-Universität zu Berlin, Berlin, Germany, 2 Institute of Biometry and Clinical Epidemiology, Charité – Universitätsmedizin Berlin, Freie Universität Berlin and Humboldt-Universität zu Berlin, Berlin, Germany, 3 Institute of Social Medicine, Epidemiology and Health Economics, Charité – Universitätsmedizin Berlin, Freie Universität Berlin and Humboldt-Universität zu Berlin, Berlin, Germany, 4 Center for Stroke Research Berlin, NeuroCure Clinical Research Center and Department of Neurology with Experimental Neurology, Charité - Universitätsmedizin Berlin, Freie Universität Berlin, Humboldt-Universität zu Berlin and Berlin Institute of Health, Berlin, Germany

* daniel.schindel@charite.de

**Data Availability Statement:** Availability of data and materials: The data associated with the paper are not publicly available due to legal restrictions imposed by the health insurance companies

## Abstract

### Background

Impairments to comprehension and production of speech (aphasia, dysarthria) and swallowing disorders (dysphagia) are common sequelae of stroke, reducing patients' quality of life and social participation. Treatment oriented on evidence-based guidelines seems likely to improve outcomes. Currently, little is known about guideline adherence in stroke aftercare for the above-mentioned sequelae. This study aims to analyse guideline adherence in the treatment of aphasia, dysarthria and dysphagia after stroke, based on suitable test parameters, and to determine factors that influence the implementation of recommended therapies.

### Methods

Six test parameters were defined, based on systematic study of guidelines for the treatment of speech impairments and swallowing disorders (e.g. comprehensive diagnostics, early initiation and continuity). Guideline adherence in treatment was tested using claims data from four statutory health insurance companies. Multivariate logistic and linear regression analyses were performed in order to test the outcomes.

### Results

4,486 stroke patients who were diagnosed with specific disorders or received speech therapy were included in the study. The median age was 78 years; the proportion of women was 55.9%. Within the first year after the stroke, 90.3% of patients were diagnosed with speech impairments and swallowing disorders. Overall, 44.1% of patients received outpatient speech and language therapy aftercare. Women were less frequently diagnosed with specific disorders (OR 0.70 [95%CI:0.55/0.88], p = 0.003) and less frequently received longer

providing the data (data contains potentially identifying patient information). The data set supporting the conclusions of this article are owned by German statutory health insurances, and are subject to strict data protection rules according to the German social security code. Therefore, the data cannot be made publicly accessible. The data we accessed is collected by health insurances when health providers bill their services towards the health insurance. The authors did not have special access privileges to the health insurances. To request the data please contact the four health insurances (versorgungsmanagement@tk.de, presse@dak.de, info@bifg.de, presse@nordost. aok.de). To fulfill the legal requirements to obtain the data, researchers must obtain permission for a specific research question from the German Federal Social Insurance Office. Additionally, researchers must conclude a contract with the statutory health insurer regarding data access. The licensee is permitted to use the data for the purpose of the research proposal. Licensees are not allowed to pass the data to a third party, or to create Software or data bases with the exception of scientific publications. For assistance in obtaining access to the data, please contact the corresponding author (DS) or institution of the corresponding author (medsoz@charite.de).

**Funding:** LS recieved funding for this study through the German Federal Ministry of Education and Research [Project NAVICARE, 01GY1601, https://www.bmbf.de/]. The funders had no role in study design, data collection and analysis, decision to publish, or preparation of the manuscript.

**Competing interests:** The authors have declared that no competing interests exist.

therapy sessions (OR 0.64 [95%CI:0.43/0.94], p = 0.022). Older age and longer hospitalization duration increased the likelihood of guideline recommendations being implemented and of earlier initiation of stroke aftercare measures.

## Conclusions

Our observations indicate deficits in the implementation of guideline recommendations in stroke aftercare. At the same time, they underscore the need for regular monitoring of implementation measures in stroke aftercare to address group-based disparities in care.

## Introduction

Annually, 795,000 strokes occur in the US [1] and approximately 262,000 in Germany [2]. Common sequelae of left hemisphere strokes are speech impairments (dysarthria), language disorders (aphasia) and swallowing disorders (dysphagia), which are treated by speech and language therapy (SLT). It has been reported that 35–52% of patients experience dysarthria [3,4], 24–41% experience aphasia and 16–44% suffer from dysphagia during the inpatient phase of stroke care [3–5]. These findings are in line with studies stating that 32% of patients receive SLT within the first years after stroke [6]. Earlier studies found 29–45% of patients with dysphagia in the acute stage, 47% in rehabilitation and 17% after 4 months [7]. Dysphagia is associated with increased mortality due to malnutrition and aspiration pneumonia, and decreased quality of life due to tube feeding or dietary restriction [3]. The high vulnerability is also apparent in higher mortality rates, levels of dependency and likelihood of placement in a nursing home [5].

Clinical practice guidelines exist for the treatment of patients with aphasia, dysarthria and dysphagia, respectively [8,9]. Guideline-adherent treatment in the acute phase has a positive effect on the patients' survival and independence after one year [10]. There is also evidence for the effectiveness of clinical guidelines in the inpatient post-acute rehabilitation of stroke: guideline adherence is associated with better recovery of physical function [11], with discharge home, with improvements on the Functional Independence Measure which includes items on communication [12], and with higher patient satisfaction [13]. However, stroke aftercare has been called a "black box" [11], as little is known about outpatient care in particular. Studies in the UK and Australia hint at limited outpatient provision and small-scale or infrequent therapies in case of chronic aphasia [14,15].

Prospective long-term studies demonstrate remaining comorbidities and complications in patients across a wide number of domains, with 47.0% of patients being in need of further stroke aftercare 41 months (study median) after stroke [6]. Figures for SLT provision are scarce, which limits the evidence base and impedes guideline development. One reason is that patients with language or speech disorders have a limited ability to provide information or even give consent. Therefore they are regularly underrepresented or excluded in research [16,17]. A claims data-based approach seems a suitable method to address that kind of selection bias and gain evidence in this hard-to-reach patient population.

Our first aim was to extract a set of testable parameters from current SLT and stroke management guidelines for aftercare treatment of aphasia, dysarthria and dysphagia. Secondly, the project aimed to analyse the current service provision of SLT in stroke aftercare and identify characteristics of those patients who are less likely to receive guideline-adherent care.

## Methods

### Patients

The sample is based on merged anonymous claims data of four statutory health insurance companies. Data were routinely collected without addressing a specific research question [18]. In total, the sample included 7,702 patients residing in a metropolitan German city who were admitted to hospital with a diagnosis of cerebrovascular disease (International Statistical Classification of Diseases and Related Health Problems German-Modification (ICD-10-GM): I60, I61, I62, I63, I64, I69, G45) in 2014. The sample used for our calculations consisted of 4,486 patients who received a diagnosis of either a specific speech disturbance (ICD-10-GM code R47, including the codes R47.0 (aphasia), R47.1 (dysarthria), and R47.8 (other and unspecified speech and language disorders)) or swallowing disorder (ICD-10-GM code R13 (dysphagia)) or any aftercare SLT. The utilisation of SLT was operationalised through the outpatient billing data from speech and language therapists. The data set comprises claims data reported for each patient one year before and one year after the initial stroke incident.

### Ethics approval and consent to participate

Ethics approval was obtained from the ethics committee of the Charité –Universitätsmedizin Berlin on 24 July 2017 (EA2/095/17, chairperson PD Dr med. E. Kaschina).

### Representativeness of data

For sample validation, the Berlin stroke registry was used [4,19]. In total, 12,006 stroke incidents were reported in the reference year 2014 [19]. Our sample linked four statutory health insurance funds, covering 64% of all events. The risk of unobserved selection bias is therefore reduced [20]. Our sample is older than the median age (78 vs. 75 years in the registry). The gender distribution differs, with women being over-represented in our sample (56.0% vs. 48.6% in the registry).

### Guideline parameters to test for adherence

19 international guidelines for stroke management existed as of 2017 [8,9] which also contain recommendations for the rehabilitation of aphasia, dysarthria and dysphagia. The three highest-valued international guidelines were included in further considerations (S1 Table). In addition, aphasia, dysarthria and dysphagia recommendations were retrieved from six German-language guidelines that were previously identified through a systematic guideline appraisal [21]. Recommendations for outpatient dysphagia rehabilitation were extracted from the three German-language guidelines identified by an additional search, paralleling that described in Mandl et al. [21]. Where guideline updates existed, the latest version was used. Guideline adherence here means an observable implementation of the defined recommendations. Extraction of parameters to test adherence using claims data was conducted by three scientific researchers: an experienced speech therapist, a medical doctor, and a medical sociologist. In a first step, all concrete and specific recommendations were extracted from each guideline. Second, comparable recommendations were grouped together, and the verifiability of the results was discussed in a consensus session.

In total, five specific testable parameters for SLT provision were extracted from the guidelines (Table 1): Patients showing speech and language disorders should be treated by professional speech and language therapists (Parameter 1: speech therapists). To account for spontaneous remission and early inpatient therapy of disorders [22–24], we defined having at least two specific diagnoses of speech disturbance (ICD-10-GM code R47, including aphasia

**Table 1. Operationalization and origin of test parameters.**

| | Recommendations | Guideline of origin | Translation into measurable parameter | Coding of target variable |
|---|---|---|---|---|
| 1 | Treating speech disorders by **speech therapists** | DEGAM, ASF, RCP, SIGN | Invoice of services (therapy by logopaedist, speech therapist) for patients with at least two specific speech disorder diagnoses (ICD-10-GM codes: R13, R47) during inpatient or outpatient care | Binary variable, Group 1: 2 specific diagnoses plus SLT Group 2: 2 specific diagnoses, no SLT Excluded: patients with 0–1 specific diagnosis |
| 2 | Comprehensive **diagnosis** at the end of or after acute phase | DGN-A, AS, GAB, SIGN | Specific diagnosis codes (ICD-10-GM) of "R13" or "R47" in inpatient or outpatient setting for stroke patients receiving SLT | Binary variable, Group 1: specific diagnosis (R13, R47) Group 2: generic stroke diagnosis (G45, I60-I64) Excluded: patients with no SLT |
| 3 | Early **initiation** and **continuity** of therapy | DGN-A, AS, GAB, RCP, SIGN | Time gap from inpatient discharge (hospital, rehabilitation) to initiation and continuity of SLT aftercare (= Number of days without SLT) | Continuous variable given in days Excluded: patients with no SLT or received SLT > 90 days |
| 4 | **Duration** of 60 minutes per session | DGN-A (as 5h/week or 3x60 minutes/week), GAB, SIGN (as 2h/week) | Sessions of 30 or 45 minutes coded as "< 60 minutes", sessions of 60 minutes coded as "60 minutes" | Binary variable, Group 1: 60 minutes Group 2: < 60 minutes Excluded: patients with no or other SLT |
| 5* | High **frequency**, with at least 2 sessions per week | AS, DGN-A, GAB (early post-acute 3x/week, later post-acute 2x/week)), DEGAM | Frequency measured by calculating average number of days between first 6 SLT sessions | Continuous variable given in days Excluded: patients with no SLT, patients with only one session |
| GAS | Guideline adherence score (GAS) | See single items 1)- 4) | Additive score of single items 1)—4), *Note*: parameter 3) was previously dichotomized at median | Continuous variable, range [0–4] |

*Calculations based on reduced sample of two health insurance companies (n = 3,339). Abbreviations: Australian Stroke Foundation (ASF), Royal College of Physicians (RCP), Scottish Intercollegiate Guideline Network (SIGN), German Society of Neurology (DGN-A), German Society of Neurology (DGN-D), Aphasie Suisse (AP), Society for Aphasia Research and Treatment (GAB) and German Society for Neurotraumatology and Clinical Neurorehabilitation (DGNKN), German Society for General and Family Medicine (DEGAM), German Society for Phoniatrics and Pedaudiology (DGPP).

(ICD-10-GM code R47.0), dysarthria (ICD-10-GM code R47.1), and "other and unspecified speech and language disorders" (ICD-10-GM code R47.8)) or swallowing difficulties (dysphagia (ICD-10-GM code R13)) during inpatient or outpatient care as the prerequisite for determining a need of aftercare SLT.

The following parameters consider the quality of SLT and therefore refer exclusively to those patients who received therapy. For these patients, comprehensive diagnosis is recommended (Parameter 2: specific diagnosis). The assumption is that coding a disease presupposes diagnosis. Patients receiving SLT were divided into two groups: those with a distinct ICD-10-GM code (R13, R47), and patients with a generic diagnosis code for stroke (G45, I60-I65).

In addition, guidelines recommend early initiation and continuity of treatment (Parameter 3: continuity). The time gap between inpatient discharge from hospital or rehabilitation to initiation of aftercare speech therapy should be as short as possible. We calculated a continuous variable for the number of days without therapy.

Next, the guidelines recommend a high intensity of treatment, which we equated with longer therapy sessions (Parameter 4). The German health care system provides for therapy sessions of 30, 45 or 60 minutes. Based on the guideline recommendations, we defined a binary

variable distinguishing between shorter ($<$ 60 minutes) vs. longer (60 minutes) duration of sessions.

Another recommendation specifies higher frequencies of at least 2 sessions per week during the post-acute phase as preferable (Parameter 5: frequencies). We received individual dates for the therapies from only two health insurance companies (n = 3,339). To measure frequency, we calculated the average days between the therapy sessions (Table 1).

Finally, a total score derived from test parameters 1 to 4 was determined for the total sample (Guideline Adherence Score/GAS). The metric test parameter 3 was dichotomised at the median beforehand for this purpose. Overall, the score permits values between 0 and 4, with a higher score implying a higher level of guideline adherence (Table 1).

### Independent variables for characterization of patients

To characterise the patients, information on age [continuous], gender [male/female] and duration of health insurance [dropout date] were taken into account. In addition, a dichotomous variable was prepared to identify strokes in the year previous to the first stroke included in our data [prior]. Patients were categorised according to the type of their initial stroke in the observation period, independent of possible subsequent recurrences with other stroke types. To measure the existing co-morbidity burden, the age-adjusted Charlson Comorbidity Index (CCI) was prepared at the time of the initial stroke included in our data. The test serves to estimate morbidity and mortality of patients based on 19 prognostically relevant comorbidities [25–27]. We used the duration of hospital stay for the initial stroke in 2014 as an indication of the severity of the stroke [severity]. In this context, patients who stay for 8 or more days are classified in the severely affected group [28,29]. The selection of relevant comorbidities is based on previous work by Van den Bussche et al. [30] and on treatment experience in relation to frequent SLT for underlying morbidities that influence the therapy.

### Data analysis

To describe the data set, the frequencies in the total sample and in the groups of stroke patients with and without SLT were descriptively analysed using the chi squared test. Due to the low number of cases, Fisher's Exact Test was used to compare rehabilitation diagnoses. Ordinal and metrically scaled variables were calculated using Median and Interquartile Range (IQR), and the Mann Whitney U-test (Wilcoxon Rank Sum test) and Student's t-Test were performed to test for differences between the groups. The five test parameters and the Guideline Adherence Score are described in analogue form. Where more than two group means were to be compared, univariate factorial analysis was used.

We performed multivariate logistic and linear regression analyses in order to test whether effects persisted after controlling for covariates. The five defined test parameters plus GAS functioned as target variables. We calculated odds ratios (OR) with 95% confidence intervals (95% CI). The software package SPSS version 25.0 was used for our statistical analysis. The significance level was set at $p<0.05$.

## Results

### Sample characteristics

The study population comprised a total of 4,486 stroke patients. Of these, 90.3% were diagnosed with dysarthria, aphasia or dysphagia and 44.1% had received outpatient SLT within the first year after the initial stroke (Table 2). The proportion of women was 56.0%. The median age was 78 years.

**Table 2. Sample characteristics: SLT and No-SLT.**

|  | Total | | No-SLT | | SLT | | p-values |
|---|---|---|---|---|---|---|---|
|  | **n** | **%** | **n** | **%** | **n** | **%** |  |
| **Total analysis sample** | 4486 | *100.0* | 2508 | *55.9* | 1978 | *44.1* |  |
| Female | 2510 | *56.0* | 1374 | *54.7* | 1136 | *45.3* | *0.076* |
| Male | 1976 | *44.0* | 1134 | *57.4* | 842 | *42.6* |  |
| Age (median, IQR) | 78 | *-* | 78 | *-* | 77 | *-* | *0.061* |
| 25% | 70 | *-* | 70 | *-* | 70 | *-* |  |
| 75% | 84 | *-* | 85 | *-* | 84 | *-* |  |
| Speech disturbances, dysphagia, *ICD-10-GM codes* R47, R13[1] | 4051 | *90.3* | 2508 | *61.9* | 1543 | *38.1* |  |
| **Inpatient diagnosis (initial hospital stay)** |  |  |  |  |  |  |  |
| *Type of stroke*, *ICD-10-GM codes* |  |  |  |  |  |  |  |
| Transient ischemic attack (TIA), G45 | 719 | *16.0* | 471 | *65.5* | 248 | *34.5* | *< 0.001* |
| Intracerebral haemorrhage (ICH), I60, I61, I62 | 560 | *12.5* | 274 | *48.9* | 286 | *51.1* | *0.002* |
| Ischemic stroke (IS), I63 | 3533 | *78.8* | 1989 | *56.3* | 1544 | *43.7* | *0.310* |
| Stroke not specified, I64 | 208 | *4.6* | 120 | *57.7* | 88 | *42.3* | *0.595* |
| *Malfunction patterns with initial stroke (inpatient diagnosis)*, *ICD-10-GM codes* |  |  |  |  |  |  |  |
| Speech disturbances, dysphagia, R47, R13 | 3738 | *83.3* | 2326 | *62.2* | 1412 | *37.8* | *< 0.001* |
| Incontinence, N39, R32, R15, G95 | 1318 | *29.4* | 630 | *47.8* | 688 | *52.2* | *< 0.001* |
| Disorders of gait and mobility, R26 | 676 | *15.1* | 365 | *54.0* | 311 | *46.0* | *0.277* |
| Paralysis, G81, G82, G83 | 2898 | *64.6* | 1550 | *53.5* | 1348 | *46.5* | *< 0.001* |
| *Comorbidities*, *ICD-10-GM codes* |  |  |  |  |  |  |  |
| Dementia, F00-F03, F05.1, G30, G31, R54 | 487 | *10.9* | 278 | *57.1* | 209 | *42.9* | *0.580* |
| Depression, F32-F33 | 416 | *9.3* | 175 | *42.1* | 241 | *57.9* | *< 0.001* |
| Parkinson, G20-G22 | 111 | *2.5* | 46 | *41.4* | 65 | *58.6* | *0.002* |
| Migraine, G43,G44 | 43 | *1.0* | 26 | *60.5* | 17 | *39.5* | *0.545* |
| Insomnia, G47,F51 | 89 | *2.0* | 39 | *43.8* | 50 | *56.2* | *0.020* |
| **Inpatient condition** |  |  |  |  |  |  |  |
| Previous stroke (n) | 267 | *6.0* | 123 | *46.1* | 144 | *53.9* | *0.001* |
| Days in hospital (median, IQR) | 11.0 | *-* | 9.0 | *-* | 19.0 | *-* | *< 0.001* |
| 25% | 6.0 | *-* | 5.0 | *-* | 7.0 | *-* |  |
| 75% | 28.0 | *-* | 23.0 | *-* | 34.0 | *-* |  |
| *Severity (n)* |  |  |  |  |  |  | *< 0.001* |
| Less affected | 1616 | *36.0* | 1094 | *67.7* | 522 | *32.3* |  |
| Severely affected | 2870 | *64.0* | 1414 | *49.3* | 1456 | *50.7* |  |
| *Charlson Index (age adjusted) (median)* | 7.0 | *-* | 7.0 | *-* | 7.0 | *-* | *0.006* |
| 25% | 5.0 | *-* | 5.0 | *-* | 6.0 | *-* |  |
| 75% | 9.0 | *-* | 9.0 | *-* | 9.0 | *-* |  |
| Drop out (12 months after discharge) (N = 4167)* | 558 | *12.4* | 352 | *63.1* | 206 | *36.9* | *< 0.001* |
| **Utilization rehabilitation services (first stay)** |  |  |  |  |  |  |  |
| Rehabilitation treatment (n) | 1159 | *25.8* | 503 | *43.4* | 656 | *56.6* | *< 0.001* |
| Rehabilitation treatment (diagnosis I60-I69) | 880 | *19.6* | 378 | *43.0* | 502 | *57.0* | *0.656* |
| Days in rehabilitation (median, IQR) | 28.0 | *-* | 26.0 | *-* | 31.0 | *-* | *< 0.001* |
| 25% | 20.0 | *-* | 20.0 | *-* | 20.0 | *-* |  |
| 75% | 44.0 | *-* | 38.0 | *-* | 49.0 | *-* |  |
| Speech disturbances, dysphagia, *ICD-10-GM codes* R47, R13 | 7 | *0.2* | 5 | *71.4* | 2 | *28.6* | *0.250* |
| Incontinence, N39, R32, R15, G95 | 4 | *0.1* | 1 | *25.0* | 3 | *75.0* | *0.638* |
| Disorders of gait and mobility, R26 | 4 | *0.1* | 2 | *50.0* | 2 | *50.0* | *1.000* |
| Paralysis, G81, G82, G83 | 18 | *0.4* | 6 | *33.3* | 12 | *66.7* | *0.476* |

*(Continued)*

**Table 2.** (Continued)

| | Total | | No-SLT | | SLT | | *p-values* |
|---|---|---|---|---|---|---|---|
| | **n** | **%** | **n** | **%** | **n** | **%** | |
| **Utilization outpatient medical care/diagnosis,** *ICD-10-GM codes* | | | | | | | |
| Contact to outpatient physician | 4157 | 92.7 | 2181 | 52.5 | 1976 | 47.5 | *< 0.001* |
| Stroke diagnosis, G45, I60-I69 | 2592 | 57.8 | 1272 | 49.1 | 1320 | 50.9 | *0.001* |
| Speech disturbances, dysphagia, R47, R13 | 886 | 19.8 | 354 | 40.0 | 532 | 60.0 | *0.001* |
| Incontinence, N39, R32, R15, G95 | 656 | 14.6 | 269 | 41.0 | 387 | 59.0 | *0.001* |
| Disorders of gait and mobility, R26 | 387 | 8.6 | 162 | 41.9 | 225 | 58.1 | *0.001* |
| Paralysis, G81, G82, G83 | 1015 | 22.6 | 372 | 36.7 | 643 | 63.3 | *0.001* |
| Comorbidities, *ICD-10-GM codes* | | | | | | | |
| Dementia, F00-F03, F05.1, G30, G31, R54 | 492 | 11.0 | 258 | 52.4 | 234 | 47.6 | *0.880* |
| Depression, F32-F33 | 605 | 13.5 | 288 | 47.6 | 317 | 52.4 | *0.016* |
| Parkinson, G20-G22 | 91 | 2.0 | 37 | 40.7 | 54 | 59.3 | *0.027* |
| Migraine, G43, G44 | 60 | 1.3 | 32 | 53.3 | 28 | 46.7 | *0.850* |
| Insomnia, G47, F51 | 188 | 4.2 | 97 | 51.6 | 91 | 48.4 | *0.883* |
| **Utilization of further outpatient therapeutic care** | | | | | | | |
| Physical therapy | 2470 | 55.1 | 798 | 32.3 | 1672 | 67.7 | *< 0.001* |
| Occupational therapy | 995 | 22.2 | 201 | 20.2 | 794 | 79.8 | *< 0.001* |

SLT = patients with a stroke who received outpatient speech and language therapy (SLT), no SLT = patients with a stroke without outpatient SLT.

*Note: Subsample used, only two health insurances included.

[1] Number of patients diagnosed at least once with dysarthria (ICD-10-GM code R47.1), aphasia (ICD-10-Code R47.0), other and unspecified speech disturbances (ICD-10-GM R47.8) or dysphagia (ICD-10-GM code R13) within the first year after initial stroke event.

## Patients with SLT

Less than half of the patients included in the study received aftercare SLT (44.1%, Table 2). Factors significantly associated with receiving SLT were a haemorrhagic stroke, higher comorbidity burden (Charlson Index, CCI), secondary diagnoses of paralysis, depression, Parkinson's, insomnia, incontinence and a long initial hospital stay (severity) (Table 2). Patients following transient ischemic attacks (TIA) less often received outpatient SLT. In addition, patients who were in inpatient rehabilitation after stroke were more likely to get SLT aftercare. With increasing rehabilitation duration, the likelihood of receiving SLT also increased.

## Description of test parameters

Women less frequently received specific speech, language or swallowing disorder diagnoses and less frequently had long treatment sessions compared to men. Speech therapists were less often involved in older patients' SLT aftercare. If SLT was applied, the treatment started earlier with increasing patient age. Similar observations were made for patients who had had a stroke in the previous year. Patients with a high Charlson Index score were shown to receive less continuity of care and shorter therapy duration (Table 3). On average, patients received one SLT session per week.

## Speech therapists (Parameter 1)

Model 1 shows, while controlling for additional influencing factors, that in addition to a stroke in the previous year (OR 1.66 [95%CI: 1.06/2.6], p = 0.025), the greater severity of the stroke

**Table 3. Descriptive analysis of recommendations.**

*For categorical parameters (1, 2, 4) the second value per group is the column %. For continuous parameters (3, 5, Guideline adherence score) the second value is the median (M/m). n = absolute numbers.*

| Test parameters | | Total n | Total %/M | Female n | Female %/m | Male n | Male %/m | Sex p | 18–49 n | 18–49 %/M | 50–59 n | 50–59 %/M | 60–69 n | 60–69 %/m | 70–79 n | 70–79 %/M | 80–89 n | 80–89 %/M | >90 n | >90 %/M | Age p | Prior stroke Yes n | Yes %/M | Prior stroke No n | No %/M | Prior p |
|---|---|---|---|---|---|---|---|---|---|---|---|---|---|---|---|---|---|---|---|---|---|---|---|---|---|---|---|
| Parameter 1: Therapists | Yes | 761 | 57.6 | 416 | 57.6 | 345 | 57.6 | 0.994 | 28 | 73.7 | 50 | 59.5 | 124 | 64.6 | 258 | 57.6 | 226 | 54.3 | 75 | 52.4 | 0.045 | 66 | 66.7 | 695 | 56.9 | 0.058 |
| | No | 560 | 42.4 | 306 | 42.4 | 254 | 42.4 | | 10 | 26.3 | 34 | 40.5 | 68 | 35.4 | 190 | 42.4 | 190 | 45.7 | 68 | 47.6 | | 33 | 33.3 | 527 | 43.1 | |
| Parameter 2: Diagnosis | Yes | 1543 | 78.0 | 855 | 75.3 | 688 | 81.7 | 0.001 | 53 | 76.8 | 100 | 82.6 | 233 | 82.0 | 560 | 79.5 | 471 | 74.4 | 126 | 75.4 | 0.059 | 110 | 76.4 | 1433 | 78.1 | 0.626 |
| | No | 435 | 22.0 | 281 | 24.7 | 154 | 18.3 | | 16 | 23.2 | 21 | 17.4 | 51 | 18.0 | 144 | 20.5 | 162 | 25.6 | 41 | 24.6 | | 34 | 23.6 | 401 | 21.9 | |
| Parameter 4: Duration | Longer sessions | 120 | 6.1 | 55 | 4.9 | 65 | 7.8 | 0.010 | 8 | 12.1 | 9 | 7.5 | 11 | 4.0 | 52 | 7.5 | 31 | 5.0 | 9 | 5.4 | 0.059 | 7 | 4.9 | 113 | 6.2 | 0.530 |
| | Shorter sessions | 1832 | 93.9 | 1060 | 95.1 | 772 | 92.2 | | 58 | 87.9 | 111 | 92.5 | 267 | 96.0 | 643 | 92.5 | 595 | 95.0 | 158 | 94.6 | | 135 | 95.1 | 1697 | 93.8 | |
| Parameter 3: Continuity | | 1780 | 13 | 1022 | 13 | 758 | 13 | 0.726 | 60 | 14 | 102 | 14 | 254 | 15 | 640 | 13 | 572 | 13 | 152 | 9.5 | <0.001 | 127 | 11 | 1653 | 13 | 0.033 |
| Parameter 5: Frequency | | 695 | 6.6 | 350 | 6.6 | 345 | 6.4 | 0.114 | 29 | 4.8 | 57 | 6.8 | 123 | 7 | 232 | 5.9 | 197 | 7 | 57 | 6.6 | 0.233 | 62 | 5.55 | 633 | 6.6 | 0.723 |
| Guideline adherence score | | 702 | 3 | 380 | 3 | 322 | 3 | 0.186 | 23 | 3 | 42 | 2 | 111 | 2 | 240 | 3 | 212 | 3 | 74 | 3 | 0.008 | 60 | 3 | 642 | 3 | 0.739 |

| Test parameters | | Total n | Total %/M | Severity 0–7 days n | 0–7 days %/m | Severity >8 days n | >8 days %/M | Severity p | Charlson [Mean / 0–6 points n] | Charlson [SD / >6 points n] | Charlson p | I60–I62 n | I60–I62 %/m | I60–I62 p | I63 n | I63 %/m | I63 p | I64 n | I64 %/m | I64 p | G45 n | G45 %/m | G45 p |
|---|---|---|---|---|---|---|---|---|---|---|---|---|---|---|---|---|---|---|---|---|---|---|---|---|
| Parameter 1: Therapists | Yes | 761 | 57.6 | 151 | 46.5 | 610 | 61.2 | 0.001 | 7.39 | 2.61 | 0.527 | 100 | 65.8 | 0.030 | 633 | 57.2 | 0.476 | 31 | 48.4 | 0.128 | 57 | 45.2 | 0.003 |
| | No | 560 | 42.4 | 174 | 53.5 | 386 | 38.8 | | 7.49 | 2.6 | | 52 | 34.2 | | 474 | 42.8 | | 33 | 51.6 | | 69 | 54.8 | |
| Parameter 2: Diagnosis | Yes | 1543 | 78.0 | 340 | 65.1 | 1203 | 82.6 | 0.001 | 7.32 | 2.56 | 0.023 | 206 | 79.8 | 0.445 | 1254 | 81.2 | 0.001 | 67 | 76.1 | 0.665 | 145 | 58.5 | 0.001 |
| | No | 435 | 22.0 | 182 | 34.9 | 253 | 17.4 | | 6.98 | 2.47 | | 52 | 20.2 | | 290 | 18.8 | | 21 | 23.9 | | 103 | 41.5 | |
| Parameter 4: Duration | Longer sessions | 120 | 6.1 | 25 | 4.8 | 95 | 6.6 | 0.138 | 6.77 | 2.49 | 0.022 | 17 | 6.8 | 0.633 | 101 | 6.6 | 0.109 | 5 | 5.7 | 0.874 | 8 | 3.2 | 0.042 |
| | Shorter sessions | 1832 | 93.9 | 495 | 95.2 | 1337 | 93.4 | | 7.28 | 2.54 | | 232 | 93.2 | | 1428 | 93.4 | | 82 | 94.3 | | 239 | 96.8 | |
| Parameter 3: Continuity | | 1780 | 13 | 454 | 13 | 1326 | 13 | 0.001 | 697 / 14 (0–6) · 1083 / 13 (>6) | | 0.540 | 236 | 14.5 | 0.808 | 1402 | 13 | 0.144 | 80 | 16 | 0.175 | 206 | 14 | 0.207 |
| Parameter 5: Frequency | | 695 | 6.6 | 156 | 7 | 539 | 6.4 | 0.006 | 294 / 6.4 (0–6) · 401 / 6.6 (>6) | | 0.503 | 98 | 7 | 0.660 | 564 | 6.6 | 0.802 | 39 | 7 | 0.721 | 53 | 8 | 0.001 |
| Guideline adherence score | | 702 | 3 | 133 | 3 | 569 | 3 | 0.827 | 252 / 3 (0–6) · 450 / 3 (>6) | | 0.738 | 89 | 3 | 0.944 | 590 | 3 | 0.301 | 30 | 2.5 | 0.442 | 48 | 2.5 | 0.853 |

Note on order of parameters: Parameters 1,2,4 are categorial variables; Parameters 3,5, GAS are continuous variables; Charlson Index (age adjusted); p = p-values; n = absolute numbers; m = median; Type of stroke: Each type included as binary variable (yes/no).

(OR 1.57 [95%CI: 1.18/2.08], p = 0.002) and also the presence of paralysis (OR 1.17 [95%CI: 1.08/1.3], p<0.001) increase the chance of a speech therapist giving treatment (Table 4).

## Specific diagnosis (Parameter 2)

According to the multivariate model, female stroke patients who received speech therapy had a lower chance of receiving a specific impairment diagnosis than male patients (OR 0.70 [95% CI: 0.55/0.88], p = 0.003) (Table 4). Equally, the predicted chances of receiving a specific diagnosis were lower for patients with transient ischemic attack compared to other types of stroke. Patients with ischemic stroke, more severely affected patients, and those with paralysis had greater chances of a specific impairment diagnosis. The likelihood of a specific disorder diagnosis was also increased for patients with a post-stroke depression (OR 1.42 [95%CI: 0.95/2.26], p = 0.084).

## Continuity (Parameter 3)

Increasing age, a stroke event in the previous year, and the occurrence of paralysis increased the likelihood of receiving earlier SLT and therefore more continuous aftercare after discharge from an acute hospital or inpatient rehabilitation centre (p>0.034, Table 5).

## Duration of sessions (Parameter 4)

In the multivariate model, being female (OR 0.64 [95%CI: 0.43/0.94],p = 0.022) and a higher Charlson Index (OR 0.89 [95%CI: 0.80/0.99], p = 0.032) were associated with a reduced chance of receiving longer therapy sessions of 60 minutes (Table 4).

## Frequencies (Parameter 5)

Higher frequencies in therapies were observed for patients with more severe stroke. In contrast, less frequent SLT was associated patients with a TIA as initial stroke in the observation period (Table 5).

## Guideline adherence score (GAS)

Higher probabilities of simultaneous implementation of several guideline recommendations were to be expected with increasing patient age (Beta 0.105, p = 0.03). Other predictors that made guideline adherence more or less likely were not identified (Table 5).

## Discussion

The findings shown provide important initial indications of guideline adherence in stroke aftercare of dysarthria, aphasia and dysphagia. As measured by the defined test parameters, we observed some disadvantageous deviations from guideline recommendations in the case of female patients, patients with severe stroke (as captured by hospitalization duration) and higher rates of comorbidities. Further, women were less likely to receive specific disorder diagnoses and longer therapy sessions. In the case of younger patients, more time elapsed before the take-up of aftercare speech therapies. In addition, patients of older age were less likely to use SLT aftercare than younger patients. A deeper look into the data shows that the implementation of recommendations is not necessarily related to specific patient groups per se. While some recommendations are implemented well, others for the same group might appear to be in great need of improvement.

**Table 4. Multivariate logistic models.** Hierarchical logistic regression models (backwards with likelihood-ratio-statistics).

| | Parameter 1 — Speech therapist (n = 1321) | | | | | | | | Parameter 2 — Specific diagnostic (n = 1978) | | | | | | | | Parameter 4 — Duration of session (n = 1952) | | | | | | | |
|---|---|---|---|---|---|---|---|---|---|---|---|---|---|---|---|---|---|---|---|---|---|---|---|---|
| | Regression coefficient B | Standard error | Wald | df | Sig. | Exp (B) | 95% CI Lower bound | Upper bound | Regression coefficient B | Standard error | Wald | df | Sig. | Exp (B) | 95% CI Lower bound | Upper bound | Regression coefficient B | Standard error | Wald | df | Sig. | Exp (B) | 95% CI Lower bound | Upper bound |
| Age | -0.012 | 0.006 | 3.611 | 1 | 0.057 | 0.99 | 0.976 | 1.0004 | -0.004 | 0.006 | 0.445 | 1 | 0.505 | 1.00 | 0.984 | 1.008 | 0.003 | 0.01 | 0.094 | 1 | 0.759 | 1.00 | 0.984 | 1,023 |
| Sex | 0.148 | 0.12 | 1.517 | 1 | 0.218 | 1.16 | 0.916 | 1.467 | -0.364 | 0.121 | 9.079 | 1 | 0.003 | **0.70** | 0.549 | 0.881 | -0.451 | 0.196 | 5.286 | 1 | 0.022 | **0.64** | 0.434 | 0.936 |
| CCI_adj | -0.037 | 0.029 | 1.634 | 1 | 0.201 | 0.96 | 0.91 | 1.02 | -0.027 | 0.03 | 0.823 | 1 | 0.364 | 0.97 | 0.918 | 1.032 | -0.115 | 0.054 | 4.599 | 1 | 0.032 | **0.89** | 0.803 | 0.99 |
| Prior | 0.506 | 0.226 | 4.99 | 1 | 0.025 | **1.66** | 1.064 | 2.583 | -0.033 | 0.214 | 0.024 | 1 | 0.878 | 0.97 | 0.636 | 1.472 | -0.154 | 0.403 | 0.146 | 1 | 0.702 | 0.86 | 0.389 | 1.889 |
| Severity | 0.448 | 0.144 | 9.665 | 1 | 0.002 | **1.57** | 1.18 | 2.077 | 0.587 | 0.131 | 20.11 | 1 | >0.001 | **1.80** | 1.392 | 2.326 | 0.423 | 0.249 | 2.88 | 1 | 0.09 | 1.53 | 0.937 | 2.487 |
| Ischemic stroke, I63 | -0.159 | 0.166 | 0.924 | 1 | 0.337 | 0.85 | 0.617 | 1.18 | 0.503 | 0.142 | 12.65 | 1 | >0.001 | **1.65** | 1.254 | 2.183 | 0.349 | 0.272 | 1.646 | 1 | 0.199 | 1.42 | 0.832 | 2.418 |
| Transient ischemic attack, G45 | -0.36 | 0.207 | 3.008 | 1 | 0.083 | 0.70 | 0.465 | 1.048 | -0.41 | 0.172 | 5.663 | 1 | 0.017 | **0.66** | 0.473 | 0.93 | -0.479 | 0.404 | 1.407 | 1 | 0.236 | 0.62 | 0.281 | 1.367 |
| Paralysis, G81-83 | 0.16 | 0.044 | 13.52 | 1 | >0.001 | **1.17** | 1.078 | 1.279 | 0.276 | 0.054 | 26.04 | 1 | >0.001 | **1.32** | 1.185 | 1.466 | 0.004 | 0.061 | 0.004 | 1 | 0.951 | 1.00 | 0.89 | 1.132 |
| Dementia, F00-F03, F05.1, G30, G31, R54 | -0.188 | 0.172 | 1.193 | 1 | 0.275 | 0.83 | 0.592 | 1.161 | 0.026 | 0.188 | 0.019 | 1 | 0.89 | 1.03 | 0.71 | 1.483 | 0.306 | 0.303 | 1.02 | 1 | 0.313 | 1.36 | 0.75 | 2.461 |
| Depression: F32-F33 | 0.078 | 0.176 | 0.193 | 1 | 0.66 | 1.08 | 0.765 | 1.527 | 0.352 | 0.204 | 2.979 | 1 | 0.084 | 1.42 | 0.953 | 2.121 | -0.471 | 0.346 | 1.856 | 1 | 0.173 | 0.62 | 0.317 | 1,23 |
| Migraine: G43, G44 | 0.385 | 0.574 | 0.449 | 1 | 0.503 | 1.47 | 0.477 | 4.522 | -0.295 | 0.567 | 0.27 | 1 | 0.603 | 0.75 | 0.245 | 2.263 | -0.035 | 1.053 | 0.001 | 1 | 0.973 | 0.97 | 0.123 | 7.602 |
| Insomnia: G47, F51 | 0.102 | 0.459 | 0.05 | 1 | 0.824 | 1.11 | 0.45 | 2.726 | -0.275 | 0.376 | 0.536 | 1 | 0.464 | 0.76 | 0.364 | 1.586 | -0.031 | 0.613 | 0.003 | 1 | 0.959 | 0.97 | 0.292 | 3.22 |
| Parkinson's disease: G20-G22 | 0.55 | 0.363 | 2.297 | 1 | 0.13 | 1.73 | 0.851 | 3.528 | -0.091 | 0.311 | 0.085 | 1 | 0.77 | 0.91 | 0.496 | 1.681 | -0.313 | 0.605 | 0.267 | 1 | 0.605 | 0.73 | 0.223 | 2.396 |
| Constant | 0.788 | 0.42 | 3.522 | 1 | 0.061 | 2.20 | | | 1.263 | 0.419 | 9.1 | 1 | 0.003 | 3.54 | | | -1.993 | 0.667 | 8.93 | 1 | 0.003 | 0.14 | | |

CCI_adj = Charlson Index (age adjusted), prior = stroke in previous year.

**Table 5. Multivariate logistic models.** Linear regression models (for continuous dependent variables).

| | Parameter 3 | | | | | | | Parameter 5 | | | | | | | Guideline adherence score | | | | | | |
|---|---|---|---|---|---|---|---|---|---|---|---|---|---|---|---|---|---|---|---|---|---|
| | Continuity n = 1780 | | | | | 95% Confidence interval for EXP (B) | | Frequency n = 695 | | | | | 95% Confidence interval for EXP (B) | | GAS n = 702 | | | | | 95% Confidence interval for EXP (B) | |
| | Regression coefficient B | Standard error | Beta | T | Sig. | Lower bound | Upper bound | Regression coefficient B | Standard error | Beta | T | Sig. | Lower bound | Upper bound | Regression coefficient B | Standard error | Beta | T | Sig. | Lower bound | Upper bound |
| Age | -0.105 | 0.049 | **-0.063** | -2.13 | 0.034 | -0.202 | -0.008 | 0.012 | 0.013 | 0.046 | 0.934 | 0.351 | -0.013 | 0.037 | 0.005 | 0.002 | **0.105** | 2.169 | 0.03 | 0 | 0.009 |
| Sex | 0.236 | 0.971 | 0.006 | 0.243 | 0.808 | -1.668 | 2.14 | 0.371 | 0.264 | 0.055 | 1.407 | 0.16 | -0.147 | 0.888 | 0.027 | 0.044 | 0.024 | 0.616 | 0.538 | -0.06 | 0.115 |
| CCI_adj | -0.198 | 0.24 | -0.026 | -0.83 | 0.408 | -0.668 | 0.271 | 0.029 | 0.066 | 0.023 | 0.432 | 0.666 | -0.102 | 0.159 | -0.005 | 0.011 | -0.025 | -0.5 | 0.617 | -0.027 | 0.016 |
| Prior | -3.964 | 1.812 | **-0.052** | -2.19 | 0.029 | -7.519 | -0.409 | -0.206 | 0.453 | -0.017 | -0.455 | 0.649 | -1.096 | 0.684 | 0.025 | 0.077 | 0.013 | 0.331 | 0.741 | -0.125 | 0.176 |
| Severity | 0.864 | 1.175 | 0.019 | 0.736 | 0.462 | -1.44 | 3.169 | -0.696 | 0.346 | **-0.086** | -2.011 | 0.045 | -1.376 | -0.016 | -0.018 | 0.059 | -0.013 | -0.306 | 0.76 | -0.134 | 0.098 |
| Ischemic stroke, I63 | -0.841 | 1.234 | -0.017 | -0.68 | 0.496 | -3.261 | 1.579 | 0.036 | 0.337 | 0.004 | 0.107 | 0.915 | -0.626 | 0.698 | 0.045 | 0.06 | 0.029 | 0.751 | 0.453 | -0.073 | 0.163 |
| Transient ischemic attack, G45 | 1.266 | 1.639 | 0.021 | 0.772 | 0.44 | -1.949 | 4.481 | 1.324 | 0.524 | **0.104** | 2.527 | 0.012 | 0.295 | 2.353 | -0.028 | 0.089 | -0.013 | -0.317 | 0.752 | -0.203 | 0.146 |
| Paralysis, G81-83 | -0.734 | 0.316 | **-0.061** | -2.32 | 0.02 | -1.354 | -0.114 | -0.009 | 0.073 | -0.005 | -0.124 | 0.901 | -0.153 | 0.135 | -0.001 | 0.012 | -0.002 | -0.05 | 0.96 | -0.025 | 0.024 |
| Dementia, F00-F03, F05.1, G30, G31, R54 | -0.841 | 1.582 | -0.013 | -0.53 | 0.595 | -3.943 | 2.261 | -0.139 | 0.42 | -0.013 | -0.331 | 0.741 | -0.963 | 0.685 | 0.031 | 0.067 | 0.018 | 0.458 | 0.647 | -0.1 | 0.161 |
| Depression: F32-F33 | -0.791 | 1.448 | -0.013 | -0.55 | 0.585 | -3.632 | 2.05 | -0.09 | 0.375 | -0.009 | -0.24 | 0.811 | -0.827 | 0.647 | -0.029 | 0.061 | -0.018 | -0.467 | 0.64 | -0.149 | 0.091 |
| Migraine: G43, G44 | -5.009 | 5.272 | -0.022 | -0.95 | 0.342 | -15.349 | 5.331 | -0.193 | 1.522 | -0.005 | -0.127 | 0.899 | -3.18 | 2.795 | 0.015 | 0.214 | 0.003 | 0.07 | 0.945 | -0.406 | 0.436 |
| Insomnia: G47, F51 | 0.92 | 2.932 | 0.007 | 0.314 | 0.754 | -4.831 | 6.67 | 0.664 | 0.816 | 0.031 | 0.814 | 0.416 | -0.937 | 2.266 | -0.065 | 0.155 | -0.016 | -0.419 | 0.676 | -0.369 | 0.24 |
| Parkinson's disease: G20-G22 | 4.247 | 2.61 | 0.039 | 1.627 | 0.104 | -0.872 | 9.366 | -1.074 | 0.709 | -0.058 | -1.515 | 0.13 | -2.466 | 0.318 | -0.078 | 0.113 | -0.026 | -0.686 | 0.493 | -0.301 | 0.145 |
| Constant | 30.717 | 3.403 | | 9.027 | <0.001 | 24.043 | 37.392 | 5.516 | 0.868 | | 6.352 | <0.001 | 3.811 | 7.221 | 2.188 | 0.153 | | 14.315 | <0.001 | 1.888 | 2.489 |

CCI_adj = Charlson Index (age adjusted), Prior = stroke in previous year.

## Sample characteristics

As we have shown, patients who receive SLT are often in a poorer state of health and have a higher comorbidity burden than patients with no SLT. For example, our findings confirm observations of previous studies that report frequent occurrence of dementia disorders and depression after strokes [31,32]. In our sample, the group receiving SLT was also disproportionally often diagnosed with insomnia. The comorbidities mentioned may make it more difficult to implement guideline recommendations, because patients' ability to cooperate may be reduced due to the progressive course of dementia or their willingness to cooperate may be impeded due to low spirits and a depressive state [33].

## Speech therapists

The majority of patients with multiple diagnosis of specific speech, language, and swallowing impairments receives aftercare through speech and language therapists. Previous studies and registers reported high inpatient therapy quotas, showing that as many as over 90% of patients received comprehensive testing and early application of SLT during their inpatient stay [4,5]. A large-scale British study reported that 77% of patients required SLT, and this was also provided for 98% of these patients during their stay in hospital [5]. However, one patient in four was discharged from hospital to their homes with impairments and very little is known about aftercare utilization among this group [5]. In an earlier study, Code et al. observed a decline in the utilization of SLT after discharge compared with treatment in hospital and rehabilitation centres [14]. In this study, the higher level of guideline adherence observed in aftercare in cases of severe stroke, paralysis and the occurrence of stroke in the previous year pointed to better implementation of the guideline recommendations in the case of highly vulnerable groups. Patients with less severe stroke or following an initial stroke less frequently received SLT aftercare, or made use of such a therapy, despite a relevant diagnosis. In this context we should bear in mind that aphasia has been found to have a higher impact on health-related quality of life than cancer and Alzheimer's disease [34], and, unsurprisingly, both aphasia and dysarthria have a negative impact on social participation [34–36]. A possible explanation might be competing priorities after discharge where in most cases, patients have to organize their complex care alone. A current study claims that older stroke survivors prioritise improving their balance and walking problems above aphasia or speech difficulties, which might explain the reduced utilization of aftercare SLT [37]. Studies show that even after leaving hospital, the majority of stroke patients still demonstrate a comprehensive need for treatment that is largely not fulfilled [6]. Contributing structural factors include the fact that organising aftercare can be overtaxing or frustrating [38] and the shortage of treatment places due to the increasing lack of trained staff in this professional field [39]. Multiple factors such as social determinants (lack of transportation, access to community service, financial situation), social support (family support, community support) and issues within the health system (disconnect between services and sectors) mean that the group of stroke patients, usually older people, must be viewed as particularly complex, and individual support needs can vary greatly [40]. The failure to address these disorders through specialists after inpatient discharge needs further examination.

## Specific diagnosis

A prerequisite for appropriate treatment is a prior comprehensive diagnosis resulting in the provision of diagnoses that are as specific as possible. Specific diagnoses have been made beforehand for the majority of patients who received SLT aftercare. It is striking, however, that the available data show a comparatively lower level of specific diagnoses for female stroke

patients. A range of gender-related differences in care are documented even at an early stage in intensive care [29,41,42]. For example, female patients more frequently report atypical symptoms, resulting in a less prompt stroke diagnosis that comes too late for thrombolytic therapy [42]. Undiagnosed and untreated pre-existing conditions, such as high blood pressure, occur more often in the group of women patients, also making them less suitable for thrombolytic therapy [42]. On the other hand, older age when the stroke occurs, linked with more frequently living alone, are associated with slower awareness of symptoms and later arrival in hospital. As a result, worse functional outcomes and more frequent impairments [41], lower quality of life [43] and higher levels of hospital mortality [29] are observed among female patients. There is often a need for post-inpatient therapy [6], but lower social status and smaller social networks seem to make this more difficult to address [17]. To explain the lower level of guideline adherence observed for female patients in relation to aftercare SLT, supplementary primary data are required to characterise them in more detail.

## Early initiation and continuity

One aspect of guideline-adherent therapy after stroke is early therapy initiation for stroke patients after hospital discharge. We found that half of the patients began aftercare within just under two weeks, with older and severely affected patients tending to start treatment faster. A study from New Zealand found comparable mean delays of 14 days until SLT initiation [44]. Minorities and persons with inadequate health insurance were less likely to receive SLT within this period [45]. Other studies stated that the initiation of therapy for the majority of patients took place after 6 weeks, with patients being particularly dissatisfied with the low amount of outpatient therapy provision [46]. In line with our results, an earlier study found that patients with aphasia who needed domiciliary visits received less therapy and at a later stage after hospital discharge than their more mobile peers [47]. Small social networks might complicate the organization of early initiation and continuity of care [17,41]. The way service provision and claims are organised in the German health care system may help explain the quicker initiation of outpatient aftercare in Germany compared to the situation in other countries. Patients with statutory health insurance usually receive a prescription for outpatient speech and language therapy from their GP. The prescription expires if the relevant outpatient aftercare treatment does not begin within 14 days. Other possible reasons behind the relatively rapid initiation of aftercare in Germany are local supply advantages due to the urban location of the study and the general obligation to hold health insurance in Germany.

## Duration of sessions

The majority of patients receives shorter therapy sessions than recommended in the guidelines. According to the guidelines, therapy should be "as tolerated and feasible" according to the patients' state of health [48]. This seems to provide a possible explanation for the lower likelihood of longer therapy sessions shown in our data for the group of patients with a higher comorbidity burden. We found no explanation as to why the majority of probably more robust female patients also receive shorter therapy sessions. International studies also report that therapy sessions in practice are shorter than recommended in clinical guidelines [5]. Both frequency and duration of sessions might be improved using telemedicine options. Preliminary studies show high levels of participation in telemedicine provision among the group of women identified as vulnerable, due to the expectation of shorter waiting times, and more frequently report greater patient satisfaction [49,50].

### Frequencies

A further recommendation found in different guidelines was that SLT should be provided with high frequency of at least 2 sessions per week. Our results indicate a lower frequency and therefore inadequate provision. Code et al. reported equivalent findings for the UK with an average of 1–1.5 hours/week treatment for chronically aphasic people attending aftercare services [14]. Low frequencies of aftercare aphasia therapy have been reported previously [15,47,51–54]. Patients needing domiciliary visits [47], patients in aged care, and those treated in private practices [54] were most at risk of under-provision. We confirm the clear gap between practice patterns and research evidence that has previously been addressed elsewhere [53]. An opposing argument against high frequency therapy was reported in Brady et al., stating that patients more often stop attending high-intensity therapy [55].

### Guideline adherence score (GAS)

The overall view shows that under this parameter, the majority of patients who required SLT were not given care in line with the guideline recommendations. The observed "age effect" underlines the attention paid by care providers when dealing with older patients. According to that, the worse physical and mental health associated with age [43] may possibly promote guideline-adherent care.

### Guideline adherence visibility in claims data

Evidence exists that clinical guidelines can improve outcomes of treatment conducted by both medical doctors [56] and by professions allied to medicine [57]. Beside bridging the research-practice gap, clinical guidelines are meant to reduce inappropriate variation in health care provision [58], thus improving health care equity. Ever since there have been guidelines, their implementation into clinical practice and guideline adherence have been an issue [59]. To our knowledge, no study has yet explicitly and comprehensively analysed whether and to what extent the provision of aftercare SLT adheres to current guidelines regarding the rehabilitation of aphasia, dysarthria and dysphagia after stroke. The analysis of health insurance claims data has been successfully applied previously to investigate guideline adherence and the influence of patient characteristics, for instance in patients with chronic coronary heart disease and peripheral arterial disease [60,61]. There were attempts in Japan to use claims data to develop quality indicators for stroke treatment [62]. Our study contributes the usability of claims data for health care services analysis even in an aftercare setting and for hard-to-reach patients, due to their communication restrictions, while providing conceivable indicators for SLT.

### Clinical relevance

The data presented here suggest that guideline adherence in outpatient follow-up of speech, language, and swallowing disorders after stroke requires significant improvement. For example, neither the guideline-recommended therapy frequencies nor the recommended 60-minute duration of therapy sessions are frequently achieved. In addition, the interval of approximately 2 weeks between discharge from the hospital and initiation of SLT treatment measures in outpatient follow-up is too long. Less than two-thirds of patients with repeatedly diagnosed speech and language disorders during acute inpatient stay were subsequently treated by outpatient SLT. The highest adherence to guidelines was observed when a specific diagnosis (as a prerequisite for SLT) was present. Structures and processes need to be established that can ensure guideline-adherent treatment of these patients in clinical practice according to their specific needs.

## Limitations

The secondary data used for this study were originally collected for the purpose of settling accounts between health insurance companies and service providers, not for scientific purposes. They are in principle suitable to reflect the care provided–with no recruitment bias. The logic of the data origin, however, means that there is a chance that patients with relevant impairments who need SLT, but who received no diagnosis or care, were not considered. The issue of large scale undercoding of diagnoses in administrative date has previously been mentioned elsewhere [63]. The authors' approach of operationalising the severity of the stroke using the duration of the initial stay in hospital (LOS) is tried and tested; however, as an individual indicator it is rather too general. A previous need for care or pre-existing conditions are as likely to cause longer hospital stays as a severe stroke. If present, a combination of additional indicators for LOS (e.g., sequelae (hemiplegia, neurological neglect), change in nursing care (e.g., to level 3 for the first time after stroke)) is recommended [28]. Where further indicators are available, e.g., in the form of the Stroke Severity Index, they should also be taken into account [64,65]. Statements on the adequacy of treatment are only possible to a limited extent. An apparent insufficiency in care provision, for example, may reflect the patients' preferences or may be oriented on patients' endurance limits, type of therapeutic stimulus or contraindications in the patient [66]. A lower level of patient take-up may be due to improved health with a concomitant reduced requirement, or to a lack of social networks and depressive episodes. The lack of specific diagnoses should be discussed because in practice, SLT is also prescribed without a specific diagnosis, simply based on the indication of a general stroke diagnosis. The comprehensive diagnosis is then carried out by the SLT therapists, whose diagnoses are not visible in the claims data. On the other hand, patients with multiple diagnoses who receive no SLT need not necessarily be viewed as inadequately provided for. Depending on the initial severity, lesion location and lesion size, spontaneous recovery is possible [22–24].

The suggested test parameters are an attempt to make existing guideline recommendations visible in secondary data. In many cases, guidelines do not include very concrete recommendations, which made it more difficult to develop reliable test parameters. One issue is that the structure of claims data limits the detailed operationalising of test parameters and a range of guideline recommendations cannot be reflected in the available claims data. This applies, for example, to the involvement and coaching of family members, or participation in self-help groups. In addition, health insurance companies give no information about the content and quality of specific therapeutic training or the patients' subjective motivation and satisfaction. Finally, we should note that the analyses are based on data from statutory health insurance companies. Patients with private insurance—who comprise only 10% of all health insurance holders in Germany, however—are therefore not included in the sample.

## Conclusion

These findings provide important initial indications for guideline adherence in the aftercare of dysarthria, aphasia and dysphagia following stroke. The suggested test parameters and the total score represent an attempt to draw aftercare treatment of speech impairments into clearer focus for future research. Continuous monitoring of the implementation of guideline recommendations may help to systematically identify disparities in care and to optimise treatment in a targeted way. Initial research findings suggest that telemedicine provision is suitable as a supplement to previous face-to-face treatment provision in order to provide targeted treatment.

## Supporting information

**S1 Table. Overview of guidelines included in the parameter construction.**
(DOCX)

## Acknowledgments

The authors thank Christoph Poggendorf for his considerable efforts in organising and preparing the data set. In addition, we would like to thank the AOK Nordost, Techniker Krankenkasse, BARMER and DAK-Gesundheit for their cooperation and provision of anonymous health insurance data.

## Author Contributions

**Conceptualization:** Daniel Schindel, Liane Schenk.

**Data curation:** Daniel Schindel, Ralph Schilling.

**Formal analysis:** Daniel Schindel, Ralph Schilling, Andreas Meisel.

**Funding acquisition:** Liane Schenk.

**Investigation:** Daniel Schindel, Liane Schenk.

**Methodology:** Daniel Schindel, Ralph Schilling, Liane Schenk.

**Project administration:** Daniel Schindel, Liane Schenk.

**Supervision:** Liane Schenk.

**Validation:** Daniel Schindel, Lena Mandl, Ralph Schilling, Andreas Meisel, Liane Schenk.

**Writing – original draft:** Daniel Schindel, Lena Mandl, Andreas Meisel.

**Writing – review & editing:** Daniel Schindel, Lena Mandl, Ralph Schilling, Andreas Meisel, Liane Schenk.

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
