## [Decision Letter · Decision Letter 0]

21 Sep 2021

PONE-D-21-17814Guideline adherence in speech and language therapy in stroke aftercare. A health insurances claims data analysisPLOS ONE

Dear Dr. Schindel,

Thank you for submitting your manuscript to PLOS ONE. After careful consideration, we feel that it has merit but does not fully meet PLOS ONE’s publication criteria as it currently stands. Therefore, we invite you to submit a revised version of the manuscript that addresses the points raised during the review process.

Please respond to reviewers and resubmit for reconsideration.

We look forward to receiving your revised manuscript.

Kind regards,

Marie Jetté

Academic Editor

PLOS ONE

2. Please include your tables as part of your main manuscript and remove the individual files. Please note that supplementary tables (should remain/ be uploaded) as separate ""supporting information"" files

Additional Editor Comments (if provided):

Reviewers' comments:

Reviewer's Responses to Questions

**Comments to the Author**

1. Is the manuscript technically sound, and do the data support the conclusions?

Reviewer #1: Yes

Reviewer #2: Partly

2. Has the statistical analysis been performed appropriately and rigorously? 

Reviewer #1: Yes

Reviewer #2: Yes

3. Have the authors made all data underlying the findings in their manuscript fully available?

Reviewer #1: Yes

Reviewer #2: Yes

4. Is the manuscript presented in an intelligible fashion and written in standard English?

Reviewer #1: No

Reviewer #2: No

5. Review Comments to the Author

Reviewer #1: Manuscript #: PONE-D-21-17814

Title: Guideline adherence in speech and language therapy in stroke aftercare. A health insurances claims data analysis

Article type: Research Article

Summary

In this manuscript, the authors defined six test parameters to measure how well speech-language therapy (SLT) aftercare aligned with clinical practice guidelines. The authors used claims data from four health insurance companies in Germany for their dataset. The findings of this manuscript are important because they shed light on reasons that individuals following stroke do not receive SLT aftercare. The overall concept and methods of this manuscript are strong, but revisions are needed prior to publication. Overall, the manuscript is difficult to read because of awkward phrasing, grammatic errors, and missing punctuation. The authors should edit the entire manuscript to improve grammar, capitalization, punctuation, phrasing, and clarity. Appropriate capitalization is particularly lacking in the tables. In addition, the following should be addressed:

Abstract

The authors mention that six test parameters were defined. Please provide one or two examples of the test parameters so that readers get a better idea of the content (Line 30).

Introduction

1. On line 59, the authors state that there are clinical practice guidelines for the treatment of aphasia, dysarthria, and dysphagia, but no references are provided. Please provide citations for established or recommended clinical practice guidelines.

Methods

1. Is there a name of the “local stroke registry” that can be provided (Line 96)?

2. Data extraction was conducted by “three scientific researchers”, but only two are described (Line 113).

3. Unless validity was tested, please remove “valid” from this sentence: “In total, five valid and specific testable parameters for SLT provision were extracted from the guidelines.” If validity was established, please indicate the type. (Line 117)

4. In describing parameter 1, the authors state that “…we defined having at least two specific diagnoses as mandatory to determine a need of ambulatory SLT” (Line 120). This appears to contradict the fourth column in Table 1 which states that the parameter was measured by “invoice of services for patients with at least one specific speech disorder diagnosis.” Please reconcile the wording of these two descriptions of parameter 1. Also, please include which specific diagnoses were included for parameter 1 in the Methods section.

5. Please provide a citation to support this sentence, if able: “Due to spontaneous remission and early inpatient therapy of disorders, we defined having at least two specific diagnoses as mandatory to determine a need of ambulatory SLT.” (Line 120)

6. The sentence that begins with “Third, guidelines recommend…” suggests that there were two sentences earlier that began “First,…” and “Second,…”. Please revise accordingly (Line 126).

7. Table 1 states that therapy duration of 60 minutes is recommended. Please add this criterion to the Methods section (around Line 130). Also, for clarity, the therapy durations for parameter 4 should be described in the same way in the Methods section and Table 1 (e.g., group 1: > 60 minutes; group 2: <60 minutes) (Line 133).

8. Table 1 states that parameter 5 is that a high frequency of therapy is recommended, with at least two sessions per week. Please add this specification to the relevant part of the Methods section (around Line 135).

9. Did the authors create the GAS, or has it been published previously? If it is based on a published method, please provide a citation.

Results:

1. Within the Results section, please specific the number and percentage of the sample that were diagnosed with dysarthria, aphasia, and dysphagia, and reference Table 2 (Lines 173-174).

2. In accordance with person-first language, consider changing “TIA patients” to “Patients following TIA…” (Line 180). Please use person-first language throughout the manuscript. Also, please define “TIA” the first time you use the abbreviation (i.e., transient ischemic attack (TIA)).

3. Please ensure that the tense of the Results section is consistent within paragraphs (e.g., past tense).

4. Please specify when treatment is “SLT” treatment throughout the article, because individuals following stroke are often seen by multiple services (e.g., “… increase the likelihood of receiving earlier and continuous care after discharge…”) (Line 207).

5. Please reference a table for the description of findings of parameter 5 and GAS (Lines 214-220). Also, please make it clearer if the description of parameter 5 findings are hypotheses or results (Lines 214 – 216).

Discussion:

1. Please revise the sentence that starts (“A final evaluation requires…”) (Line 228) which is difficult to understand.

2. The discussion section titled “Speech therapists” needs to be revised to improve flow and coherence. Presently, it is difficult to understand the main points that the authors are communicating.

3. The sentence at the beginning of the “Specific diagnostics” discussion section is unclear: “The prerequisite for appropriate treatment is correct diagnosis, which seems to have occurred in a large majority of cases.” (Line 259). As this is a retrospective analysis of claims data, it is impossible to know if correct diagnoses were made. Please revise.

4. In the “Early onset and continuity” discussion, the authors report that half of their patients started SLT aftercare within two weeks after discharge, and mention reports from other parts of the world that have slower initiation of treatment. It might be helpful for the authors to suggest possible reasons that the German healthcare system may be able to initiate earlier services. (Line 275)

5. An article that might help support the concern described in the third sentence of the Limitations section, is: González-Fernández, M., Gardyn, M., Wyckoff, S., Ky, P. K. S., & Palmer, J. B. (2009). Validation of ICD-9 code 787.2 for identification of individuals with dysphagia from administrative databases. Dysphagia, 24(4), 398–402.

6. In the US, we have the same problem with SLT diagnoses not being represented well in claims data as you describe in Lines 344-345. I have no revision request for this point.

Table 1.

1. This is a very useful table that makes it clear how the test parameters were generated and quantified. Please add appropriate capitalization throughout the table.

2. The third recommendation “early onset and continuity” should be described more clearly (e.g., “early initiation of therapy and continuity”). It is described clearly in the text of the article, but it is not described as well in the table.

Table 2.

1. In the first column under “age (median, IQR)”, “25” and “75” should have percent signs following the numbers if these are quartiles of the IQR.

2. Please change the title to something clearer, such as “Sample Characteristics: SLT vs. No-SLT

Reviewer #2: This manuscript describes efforts to define and test the typical clinical implementation of guidelines for SLT driven care for patients who are post-stroke with diagnosed aphasia, dysarthria, and/or dysphagia. The authors reviewed international and national (German) guidelines. Six applicable guidelines were identified. Using claims data from 4,486 stroke patients, the investigators determined the degree to which care complied with guidelines and explored factors that might be associated with guideline non-compliance. Overall, the article provides important information that contributes to the emerging literature base that demonstrates that SLT care often does not adhere to guidelines, especially have patients are discharged from the acute stroke hospitalization. While many factors are adequately accounted for, stroke severity is not. Stroke severity likely plays a major role in patients receiving SLT. In this study, length of initial acute hospitalization was used as a proxy for stroke severity. There are other published markers of stroke severity which should be incorporated to allow for better interpretation and application of the results. However, even without strong stroke severity data, this manuscript calls attention to a very large issue in the field – guidelines for SLT care for the post-stroke population are generally not adhered to. Given the magnitude of the issue, a paragraph about Clinical Relevance at the end of the discussion section detailing the guidelines that are not uniformly adhered to would be helpful to calling clinician and healthcare management attention to the results.

Specific Questions/Issues:

1. Lines 120-121: It is unclear what is meant by at least 2 diagnoses. Please specify. If what is meant is stroke + aphasia, dysarthria, or dysphagia, then this makes sense. If it refers to something else, please provide further justification.

2. Lines 181-182: This sentence seems to refer to patients already or previously being treated by an SLT prior to the incident stroke (stroke that got them enrolled in the current retrospective study). If so, please provide more information.

3. Lines 184-187: It is unclear what is meant by the results in these 3 sentences. Please rephrase.

4. The discussion seems to under interpret the seriousness of the issue. Of 83% of post-stroke patients with an SLT diagnosis, only 38% of those received SLP services. Please strengthen the discussion related to this.

5. In Table 2, why are so few instances of SLT, incontinence, gait, or paralysis coded for the first stay utilization.

6. Please add a Clinical Relevance section at the end of the discussion section detailing the guidelines that are not uniformly adhered to would be helpful to calling clinician and healthcare management attention to the results.

7. Please add as a limitation the lack of direct measure of stroke severity and the issues with using LOS of first hospitalization as a proxy.

6. PLOS authors have the option to publish the peer review history of their article (what does this mean?). If published, this will include your full peer review and any attached files.

Reviewer #1: No

Reviewer #2: No

---

## [Author Response · Author response to Decision Letter 0]

15 Nov 2021

PONE-D-21-17814

Guideline adherence in speech and language therapy in stroke aftercare. A health insurance claims data analysis

PLOS ONE

Dear Professor Jetté, 

Thank you very much for reviewing our manuscript and giving us the opportunity to revise it with your help and the help of the reviewers. We discussed the comments in the authors' group and incorporated them point by point into the manuscript. We have included our answers (AW) and the revised text sections directly below the reviewers' notes

Kind regards

Dr. Daniel Schindel

Reviewer #1: Manuscript #: PONE-D-21-17814

Title: Guideline adherence in speech and language therapy in stroke aftercare. A health insurances claims data analysis

Article type: Research Article

Summary

In this manuscript, the authors defined six test parameters to measure how well speech-language therapy (SLT) aftercare aligned with clinical practice guidelines. The authors used claims data from four health insurance companies in Germany for their dataset. The findings of this manuscript are important because they shed light on reasons that individuals following stroke do not receive SLT aftercare. The overall concept and methods of this manuscript are strong, but revisions are needed prior to publication. Overall, the manuscript is difficult to read because of awkward phrasing, grammatic errors, and missing punctuation. The authors should edit the entire manuscript to improve grammar, capitalization, punctuation, phrasing, and clarity. Appropriate capitalization is particularly lacking in the tables. In addition, the following should be addressed:

Dear Reviewer, 

Thank you for your detailed appraisal of our manuscript. Your hints and thoughts were very helpful and useful for our revision of the paper. The authors discussed the points you mentioned in detail and implemented them all. 

Best wishes

The authors 

Abstract

The authors mention that six test parameters were defined. Please provide one or two examples of the test parameters so that readers get a better idea of the content (Line 30).

AW (Answer by the authors): Thank you for the comment. We added two examples in brackets.

Revised Sentence: Six test parameters were defined, based on systematic research of guidelines for the treatment of speech impairments and swallowing disorders (e.g. comprehensive diagnostic, early initiation and continuity).

Introduction

1. On line 59, the authors state that there are clinical practice guidelines for the treatment of aphasia, dysarthria, and dysphagia, but no references are provided. Please provide citations for established or recommended clinical practice guidelines.

AW: We provided citations for two very helpful papers (Rohde 2013, Shrubsole 2017) evaluating clinical guidelines for post-stroke speech and language disorders rehabilitation.

Methods

1. Is there a name of the “local stroke registry” that can be provided (Line 96)?

AW: We named the registry and added citations.

Revised Sentence: “For sample validation, the Berlin stroke registry was used [4, 19].”

2. Data extraction was conducted by “three scientific researchers”, but only two are described (Line 113).

AW: Thank you. We added the third person.

Revised Sentence: “Extraction of parameters to test adherence using claims data was conducted by three scientific researchers: an experienced speech therapist, and a medical doctor and a medical sociologist were involved.”

3. Unless validity was tested, please remove “valid” from this sentence: “In total, five valid and specific testable parameters for SLT provision were extracted from the guidelines.” If validity was established, please indicate the type. (Line 117)

AW: We removed the word “valid”.

4. In describing parameter 1, the authors state that “…we defined having at least two specific diagnoses as mandatory to determine a need of ambulatory SLT” (Line 120). This appears to contradict the fourth column in Table 1 which states that the parameter was measured by “invoice of services for patients with at least one specific speech disorder diagnosis.” Please reconcile the wording of these two descriptions of parameter 1. Also, please include which specific diagnoses were included for parameter 1 in the Methods section.

AW: Thank you very much for pointing this out. There is indeed an error in column 4 of Table 1. We changed that into “two specific diagnoses” and included specific diagnoses. The specific diagnoses were also included in the methods section.

Revised Table 1: See column 3, row 2

Revised sentence in Methods section: Due to spontaneous remission and early inpatient therapy of disorders [22-24], we defined having at least two specific diagnoses of speech disturbance (ICD-10-GM code R47 (Aphasia, Dysarthria) or swallowing difficulties (ICD-10-GM code R13 (Dysphagia)) during inpatient or outpatient care as mandatory to determine a need of aftercare SLT.

5. Please provide a citation to support this sentence, if able: “Due to spontaneous remission and early inpatient therapy of disorders, we defined having at least two specific diagnoses as mandatory to determine a need of ambulatory SLT.” (Line 120)

AW: We provided citations to support the sentence. (Sorry, citations for the occurrence of spontaneous remissions were only given in line 347 in the first version.)

Revised Sentence: “Due to spontaneous remission and early inpatient therapy of disorders [Plowman 2012, Ferro 1999, Fridriksson 2012], [….]”

6. The sentence that begins with “Third, guidelines recommend…” suggests that there were two sentences earlier that began “First,…” and “Second,…”. Please revise accordingly (Line 126).

AW: Of course. We changed the sentence and revised the “Fourth, ..” two sentences later as well.

7. Table 1 states that therapy duration of 60 minutes is recommended. Please add this criterion to the Methods section (around Line 130). Also, for clarity, the therapy durations for parameter 4 should be described in the same way in the Methods section and Table 1 (e.g., group 1: > 60 minutes; group 2: <60 minutes) (Line 133).

AW: We made the description in Table 1 more precise and rearranged the part in the Methods section.

Revised part in Table 1, column 4, row 5.

Revised sentences in Methods section: “The German health care system provides for three possible durations of therapy sessions (30, 45 and 60 minutes). Based on the guideline recommendations, we defined a binary variable distinguishing between shorter (< 60 minutes) vs. longer (60 minutes) duration of sessions.”

8. Table 1 states that parameter 5 is that a high frequency of therapy is recommended, with at least two sessions per week. Please add this specification to the relevant part of the Methods section (around Line 135).

AW: We included this specification. 

Revised sentence: “Another recommendation specifies higher frequencies of at least 2 sessions per week during the post-acute phase as preferable (Parameter 5: frequencies).”

9. Did the authors create the GAS, or has it been published previously? If it is based on a published method, please provide a citation.

AW: We developed the GAS scale ourselves. The reasoning behind it was that a score is easier to use in future research or practice than comparing five single items.

Results:

1. Within the Results section, please specific the number and percentage of the sample that were diagnosed with dysarthria, aphasia, and dysphagia, and reference Table 2 (Lines 173-174).

AW: We added the specific number of patients in Table 2 (line 9) and rearranged the sentence in the Results section. 

Revised sentences: “The study population comprised a total of 4,486 stroke patients. Of these, 90.3% were diagnosed with dysarthria, aphasia or dysphagia and 44.1%or who had received outpatient SLT within the first year after the initial stroke (Table 2). The proportion of women was 56.0%.”

2. In accordance with person-first language, consider changing “TIA patients” to “Patients following TIA…” (Line 180). Please use person-first language throughout the manuscript. Also, please define “TIA” the first time you use the abbreviation (i.e., transient ischemic attack (TIA)).

AW: We changed the wording to person-first language and included the definition of TIA. 

Revised sentence: “Patients following transient ischemic attacks (TIA) less often received outpatient SLT.”

3. Please ensure that the tense of the Results section is consistent within paragraphs (e.g., past tense).

AW: After revising the content based on the reviewers' comments, we had the manuscript checked by a professional proofreading service.

4. Please specify when treatment is “SLT” treatment throughout the article, because individuals following stroke are often seen by multiple services (e.g., “… increase the likelihood of receiving earlier and continuous care after discharge…”) (Line 207).

AW: Of course, thank you for pointing this out. We have added to the paragraph to make it clearer. 

“Increasing age, a stroke event in the previous year, and the occurrence of paralysis increased the likelihood of receiving earlier SLT and therefore a more continuous aftercare after discharge from acute hospital or inpatient rehabilitation center (p>0.034, Table 5).”

5. Please reference a table for the description of findings of parameter 5 and GAS (Lines 214-220). Also, please make it clearer if the description of parameter 5 findings are hypotheses or results (Lines 214 – 216).

AW: Thank you. We have added a reference to the corresponding tables and rephrased the paragraph on parameter 5.

Revised paragraph: “Higher frequencies in therapies were observed for patients with more severe stroke. In contrast, less frequent SLT was associated patients with a TIA as initial stroke in the observation period (Table 5).”

Discussion:

1. Please revise the sentence that starts (“A final evaluation requires…”) (Line 228) which is difficult to understand.

AW: We revised the sentence. 

Revised sentence: “A final evaluation requires a detailed look, as the designated groups are not disadvantaged across all test parameters.”

2. The discussion section titled “Speech therapists” needs to be revised to improve flow and coherence. Presently, it is difficult to understand the main points that the authors are communicating.

AW: We rephrased the paragraph, and added literature to strengthen the main points. 

New paragraph: 

“The majority of patients with a repeated diagnosis of specific speech and language impairments receives aftercare through speech and language therapists. Previous studies and registers reported high inpatient therapy quotas [4, 5] showing that as many as over 90% of patients received comprehensive testing and early application of SLT during their inpatient stay [4]. A large-scale British study reported that 77% of patients required SLT, and this was also provided for 98% of these patients during their stay in hospital [5]. However, one patient in four was discharged from hospital to their homes with impairments and very little is known about aftercare utilization among this group [5]. In an earlier study, Code et al. observed a decline in the utilization of SLT after discharge compared with treatment in hospital and rehabilitation centers [14]. In this study, the higher level of guideline adherence observed in aftercare in cases of severe stroke, paralysis and the occurrence of stroke in the previous year pointed to better implementation of the guideline recommendations in the case of highly vulnerable groups. Patients with less severe stroke or first stroke less frequently received SLT aftercare, or made use of such a therapy, despite a relevant diagnosis. In this context we should bear in mind that aphasia has been found to have a higher impact on health-related quality of life than cancer and Alzheimer’s disease [31], and, unsurprisingly, both aphasia and dysarthria have a negative impact on social participation [31-33]. A possible explanation might be competing priorities after discharge where in most cases, patients have to organize their complex care alone. A current study claims that older stroke survivors prioritise improving their balance and walking problems above aphasia or speech difficulties, which might explain the reduced utilization of aftercare SLT [34]. Studies show that even after leaving hospital, the majority of stroke patients still demonstrate a comprehensive need for treatment that is largely not fulfilled [6]. Contributing structural factors include the fact that organising aftercare can be overtaxing or frustrating [38] and the shortage of treatment places due to the increasing lack of trained staff in this professional field [39]. Multiple factors such as social determinants (lack of transportation, access to community service, financial situation), social support (family support, community support) and issues within the health system (disconnect between services and sectors) mean that the group of stroke patients, usually older people, must be viewed as particularly complex, and individual support needs can vary greatly [40]. The failure to address these disorders through specialists after inpatient discharge needs further examination.”

3. The sentence at the beginning of the “Specific diagnostics” discussion section is unclear: “The prerequisite for appropriate treatment is correct diagnosis, which seems to have occurred in a large majority of cases.” (Line 259). As this is a retrospective analysis of claims data, it is impossible to know if correct diagnoses were made. Please revise.

AW: The sentence was split into two sentences and revised. We hope that the statements are clearer now.

Revised sentences: “A prerequisite for appropriate treatment is a prior comprehensive diagnosis resulting in the provision of diagnoses that are as specific as possible. Specific diagnoses have been made beforehand for the majority of patients who received SLT.”

4. In the “Early onset and continuity” discussion, the authors report that half of their patients started SLT aftercare within two weeks after discharge, and mention reports from other parts of the world that have slower initiation of treatment. It might be helpful for the authors to suggest possible reasons that the German healthcare system may be able to initiate earlier services. (Line 275)

AW: We added a few sentences suggesting possible reasons for the early initiation of SLT. 

New sentences: “The way service provision and claims are organised in the German health care system may help explain the quicker initiation of outpatient aftercare in Germany compared to the situation in other countries. Patients with statutory health insurance usually receive a prescription for outpatient speech and language therapy from their GP. The prescription expires if the relevant outpatient aftercare treatment does not begin within 14 days. Other possible reasons behind the relatively rapid initiation of aftercare in Germany are local supply advantages due to the urban location of the study and the general obligation to hold health insurance in Germany.”

5. An article that might help support the concern described in the third sentence of the Limitations section, is: González-Fernández, M., Gardyn, M., Wyckoff, S., Ky, P. K. S., & Palmer, J. B. (2009). Validation of ICD-9 code 787.2 for identification of individuals with dysphagia from administrative databases. Dysphagia, 24(4), 398–402.

AW: Thank you for that paper! We added a sentence to refer to the problem of undercoding in administrative data.

New Sentence: “The issue of large scale undercoding of diagnoses in administrative date was previously mentioned elsewhere before [63].”

6. In the US, we have the same problem with SLT diagnoses not being represented well in claims data as you describe in Lines 344-345. I have no revision request for this point.

Table 1.

1. This is a very useful table that makes it clear how the test parameters were generated and quantified. Please add appropriate capitalization throughout the table.

AW: Thank you for the comment. We corrected the capitalization. 

2. The third recommendation “early onset and continuity” should be described more clearly (e.g., “early initiation of therapy and continuity”). It is described clearly in the text of the article, but it is not described as well in the table.

AW: We have matched the description in Table 1 to the description in the Methods section.

Table 2.

1. In the first column under “age (median, IQR)”, “25” and “75” should have percent signs following the numbers if these are quartiles of the IQR.

AW: Thank you. We added percent signs to all IQR numbers in Table 2. 

2. Please change the title to something clearer, such as “Sample Characteristics: SLT vs. No-SLT

AW: We have shortened the title.

Reviewer #2

Reviewer #2: This manuscript describes efforts to define and test the typical clinical implementation of guidelines for SLT driven care for patients who are post-stroke with diagnosed aphasia, dysarthria, and/or dysphagia. The authors reviewed international and national (German) guidelines. Six applicable guidelines were identified. Using claims data from 4,486 stroke patients, the investigators determined the degree to which care complied with guidelines and explored factors that might be associated with guideline non-compliance. Overall, the article provides important information that contributes to the emerging literature base that demonstrates that SLT care often does not adhere to guidelines, especially have patients are discharged from the acute stroke hospitalization. While many factors are adequately accounted for, stroke severity is not. Stroke severity likely plays a major role in patients receiving SLT. In this study, length of initial acute hospitalization was used as a proxy for stroke severity. There are other published markers of stroke severity which should be incorporated to allow for better interpretation and application of the results. However, even without strong stroke severity data, this manuscript calls attention to a very large issue in the field – guidelines for SLT care for the post-stroke population are generally not adhered to. Given the magnitude of the issue, a paragraph about Clinical Relevance at the end of the discussion section detailing the guidelines that are not uniformly adhered to would be helpful to calling clinician and healthcare management attention to the results.

Dear Reviewer, 

Thank you for your detailed appraisal of our manuscript. Your hints and thoughts were very helpful and useful for our revision of the paper. The authors discussed the points you mentioned in detail and implemented them as carefully as possible. We agree that statements about the severity of a stroke based solely on the duration of the hospital stay only give a limited view as this is not a specific indicator. Patient-related factors such as previous diseases or comorbidities, as well as structural and organisational conditions of care, may distort the picture. It would be desirable to include additional indicators to provide a severity score, for example [Sung 2016a, b]. Unfortunately, the claims data give no detailed information about procedures carried out (e.g. ventilation of the patient, coma, PEG tube (e.g. coded in accordance with the official classification for the encoding of operations, procedures and general medical measures)). In addition, we had no access to administrative hospital data, which regularly include the Barthel Index and modified Rankin Scale. However, we have additional indicators suggested in the literature in our descriptions of patients and guideline implementation, such as the presence of hemiplegia or hemiparesis or the Charlson Index. 

Best wishes

The authors

Specific Questions/Issues:

1. Lines 120-121: It is unclear what is meant by at least 2 diagnoses. Please specify. If what is meant is stroke + aphasia, dysarthria, or dysphagia, then this makes sense. If it refers to something else, please provide further justification.

AW (Answer by the authors): Thank you for pointing this out. We added the specific ICD-10-codes in brackets. 

New sentences: “Due to spontaneous remission and early inpatient therapy of disorders [22-24], we defined having at least two specific diagnoses of speech disturbance (ICD-10-GM code R47 (Aphasia, Dysarthria) or swallowing difficulties (ICD-10-GM code R13 (Dysphagia)) during inpatient or outpatient care as mandatory to determine a need of aftercare SLT.”

2. Lines 181-182: This sentence seems to refer to patients already or previously being treated by an SLT prior to the incident stroke (stroke that got them enrolled in the current retrospective study). If so, please provide more information.

AW: Thank you for this question. We revised the sentence. The reported results refer exclusively to the care of the patients after the initial stroke.

Revised sentence: “Patients receiving aftercare SLT were more frequently and for a longer period in rehabilitation measures after the initial stroke.”

Note: Please note that patients who have spent long periods in hospital or rehabilitation centers have generally also received SLT there. However, we were not able to reflect this in our data due to the claim procedures for inpatient care and rehabilitation. We could only view the claims for SLT in outpatient aftercare. Further, it should also be taken into account that while it is true that some of the patients had already received outpatient SLT before the stroke we defined as the initial stroke, these patients usually had a stroke in the previous year or had a different condition that requires SLT (Parkinson, dementia).

3. Lines 184-187: It is unclear what is meant by the results in these 3 sentences. Please rephrase.

AW: We rephrased the sentences as follows: 

New sentences: “Women less frequently received specific speech, language or swallowing disorder diagnoses and less frequently had long treatments sessions compared to men. Speech therapists were less often involved in older patients’ SLT aftercare. If SLT was applied, the treatment started earlier with increasing patient age.”

4. The discussion seems to under interpret the seriousness of the issue. Of 83% of post-stroke patients with an SLT diagnosis, only 38% of those received SLP services. Please strengthen the discussion related to this.

AW: Thank you for pointing this out. We have expanded the discussion. It is true that at 83%, the proportion of patients with at least one speech or language disorder diagnosis in hospital is very high. The fact that fewer than half of the patients claim outpatient SLT can only be partly explained by the occurrence of spontaneous remission, inpatient treatment of the disorder or patients setting their own priorities. Studies show that even after leaving hospital, the majority of stroke patients still demonstrate a comprehensive need for treatment that is largely not fulfilled. Contributing structural factors include the fact that organising aftercare can be overtaxing or frustrating and the shortage of treatment places due to the increasing lack of trained staff in this professional field. Multiple factors such as social determinants, social support and issues within the health system mean that the group of stroke patients, usually older people, must be viewed as particularly complex, and individual support needs can vary greatly.

New paragraph: “A possible explanation might be competing priorities after discharge where in most cases, patients have to organize their complex care alone. A current study claims that older stroke survivors prioritise improving their balance and walking problems above aphasia or speech difficulties, which might explain the reduced utilization of aftercare SLT [37]. Studies show that even after leaving hospital, the majority of stroke patients still demonstrate a comprehensive need for treatment that is largely not fulfilled [6]. Contributing structural factors include the fact that organising aftercare can be overtaxing or frustrating [38] and the shortage of treatment places due to the increasing lack of trained staff in this professional field [39]. Multiple factors such as social determinants (lack of transportation, access to community service, financial situation), social support (family support, community support) and issues within the health system (disconnect between services and sectors) mean that the group of stroke patients, usually older people, must be viewed as particularly complex, and individual support needs can vary greatly [40]. The failure to address these disorders through specialists after inpatient discharge needs further examination.”

5. In Table 2, why are so few instances of SLT, incontinence, gait, or paralysis coded for the first stay utilization.

AW: Thank you for studying Table 2 with such care. The data on incontinence and paralysis conform to the functional restrictions at the time of entry into hospital as documented in the Berlin stroke registry. The comparatively very low number of “disorders of gait and mobility” (ICD-10 code: R26) is probably due to the coding logic of the “R codes” of the ICD -10 catalogue that actually describe symptoms, not diagnoses. After discussing this point with physicians at the Berlin stroke center we can confirm that in the case of “paralysis”, the additional coding “gait disorder” is frequently not included. The undercoding is thus determined by the claims-oriented coding.

6. Please add a Clinical Relevance section at the end of the discussion section detailing the guidelines that are not uniformly adhered to would be helpful to calling clinician and healthcare management attention to the results.

AW: Thanks you for this suggestion. We have added a short paragraph giving the core points of the analysis. This enables clinician and healthcare management to identify at a glance the guideline recommendations that are not, or not adequately, implemented.

New paragraph: “Clinical relevance. 

“The data presented here suggest that guideline adherence in outpatient follow-up of speech, language, and swallowing disorders after stroke requires significant improvement. For example, neither the guideline-recommended therapy frequencies nor the recommended 60-minute duration of therapy sessions are frequently achieved. In addition, the interval of approximately 2 weeks between discharge from the hospital and initiation of SLT treatment measures in outpatient follow-up is too long. Less than two-thirds of patients with repeatedly diagnosed speech and language disorders during acute inpatient stay were subsequently treated by outpatient SLT. The highest adherence to guidelines was observed when a specific diagnosis (as a prerequisite for SLT) was present. Structures and processes need to be established that can ensure guideline-conformant treatment of these patients in clinical practice according to their specific needs."

7. Please add as a limitation the lack of direct measure of stroke severity and the issues with using LOS of first hospitalization as a proxy.

AW: As mentioned earlier, we agree that statements about the severity of a stroke based solely on the duration of the hospital stay only give a limited view as this is a non-specific indicator. Patient-related factors such as previous illnesses or comorbidities as well as structural and organisational conditions of care may distort the picture. It would be desirable to include additional indicators in our description. Unfortunately, our claims data give us no additional detailed information about procedures carried out (e.g., ventilation of the patient, coma, PEG tube (e.g., coded in accordance with the official classification for the encoding of operations, procedures and general medical measures)). In addition, we had no access to hospital administrative data, which regularly include the Barthel Index and modified Rankin Scale.

New sentences in Limitations section: “The authors’ approach of operationalising the severity of the stroke using the duration of the initial stay in hospital (LOS) is tried and tested; however, as an individual indicator it is rather too general. A previous need for care or pre-existing conditions are as likely to cause longer hospital stays as a severe stroke. If present, a combination of additional indicators for LOS (e.g., sequelae (hemiplegia, neurological neglect), change in nursing care (e.g., to level 3 for the first time after stroke)) is recommended [28]. Where further indicators are available, e.g. in the form of the Stroke Severity Index, they should also be taken into account [64, 65].”

---

## [Decision Letter · Decision Letter 1]

14 Dec 2021

PONE-D-21-17814R1Guideline adherence in speech and language therapy in stroke aftercare. A health insurance claims data analysis.PLOS ONE

Dear Dr. Schindel,

Thank you for submitting your manuscript to PLOS ONE. After careful consideration, we feel that it has merit but does not fully meet PLOS ONE’s publication criteria as it currently stands. Therefore, we invite you to submit a revised version of the manuscript that addresses the points raised during the review process.

 Specifically, please respond to Reviewer 1's revision requests. 

We look forward to receiving your revised manuscript.

Kind regards,

Marie Jetté

Academic Editor

PLOS ONE

Reviewers' comments:

Reviewer's Responses to Questions

**Comments to the Author**

1. If the authors have adequately addressed your comments raised in a previous round of review and you feel that this manuscript is now acceptable for publication, you may indicate that here to bypass the “Comments to the Author” section, enter your conflict of interest statement in the “Confidential to Editor” section, and submit your "Accept" recommendation.

Reviewer #1: All comments have been addressed

Reviewer #2: All comments have been addressed

2. Is the manuscript technically sound, and do the data support the conclusions?

Reviewer #1: Yes

Reviewer #2: (No Response)

3. Has the statistical analysis been performed appropriately and rigorously? 

Reviewer #1: Yes

Reviewer #2: (No Response)

4. Have the authors made all data underlying the findings in their manuscript fully available?

Reviewer #1: Yes

Reviewer #2: (No Response)

5. Is the manuscript presented in an intelligible fashion and written in standard English?

Reviewer #1: No

Reviewer #2: (No Response)

6. Review Comments to the Author

Reviewer #1: Manuscript #: PONE-D-21-17814R1

Title: Guideline adherence in speech and language therapy in stroke aftercare. A health insurances claims data analysis

Article type: Research Article

Summary

The authors have carefully responded to the reviewer comments and revised their manuscript accordingly. There are some additional minor concerns that have been identified (below) that could be addressed. Additionally, there are several remaining grammatical issues. It would likely improve the readability of the manuscript if it was proofread/revised in its entirety for English grammar again. If these minor issues are addressed satisfactorily, it is likely that this manuscript could be accepted.

Abstract

In the sentence that starts “Treatment oriented on medical guidelines…”, the word “medical” could be misleading, as it often refers to medication. Perhaps this could be changed to “evidence-based guidelines” or something similar.

As Reviewer 2 points out, using hospitalization duration as a proxy for stroke severity is not ideal. To make the abstract more clearly represent the data, please revise the sentence that starts “Older age and increasing severity of stroke…” by replacing “severity of stroke” with “hospitalization duration”.

The abstract seems to be missing some of the most salient findings of the paper. Please consider including in the Results section of the Abstract that of the 90.3% of post-stroke patients with an SLT diagnosis, only 44.1% received outpatient SLP services within the first year post-stroke.

Methods

In the Guideline parameters to test for adherence section: From the sentence that starts “Extraction of parameters”, please delete “were involved” from the end of the sentence (Lines 119-121). Also, please add a comma after “medical doctor”.

In the Guideline parameters to test for adherence section: Please add a colon after the sentence that starts “In total, five specific testable…” Otherwise, the sentence that follows is hard to interpret. (Lines 125-126).

In the Guideline parameters to test for adherence section: It is still difficult to interpret the sentence that includes, “we defined having at least two specific diagnoses of speech disturbance ...” (Lines 127 – 131). Perhaps you could add the specific ICD-10-GM codes for aphasia and dysarthria (R47.01?) and separate these two conditions in the sentence (e.g., “aphasia (ICD-10-GM code R47.X), dysarthria (ICD-10-GM code R47.X”)) so that the sentence is clearer. Presently, it looks like dysarthria and aphasia have the same ICD-10 code.

Results

The last sentence of the Patients with SLT subsection needs to be revised (“Patients receiving aftercare SLT were more frequently and for a longer period in rehabilitation measures after the initial stroke”). (Lines 204-205)

Discussion

In the second sentence of the first paragraph of the Discussion, it would be helpful to clarify that hospitalization duration was used as a proxy for stroke severity by putting this information in parentheses (i.e., “patients with severe stroke (as captured by hospitalization duration)”). (Lines 256 - 258).

Please make the fourth sentence of the first paragraph of the Discussion section clearer (“Younger patients needed more time for the take-up of aftercare speech therapies.”) (Line 259-260)

Please revise the last sentence of the first paragraph of the Discussion section that starts “A final evaluation…” (Lines 261-262). Perhaps you could start the sentence with, “A deeper look into the data shows that …”, or something similar that hints to the reader what will be discussed in the remainder of the Discussion.

In the first sentence of the Speech therapists subsection of the Discussion, do you mean “multiple diagnoses” rather than “repeated diagnosis”? Could you add swallowing impairments to this sentence?

In the first paragraph of the Speech therapists subsection of the Discussion, please revise “Patients with less severe stroke or first stroke” to “Patients with less severe stroke or following an initial stroke”. (Lines 285-286)

In the first paragraph of the Early Initiation and continuity subsection of the Discussion, please change “therapy resumption” in the first sentence to “therapy initiation.” (Line 324)

Tables

On Table 2, it might be helpful to revise the 7th row description from “Speech, language disorders: R47, R13” to “Dysarthria, aphasia, or dysarthria” (and add the relevant ICD-10 codes). Please include the raw number and percentage of the population for each of the three codes in the table (e.g., How many individuals had dysphagia?).

Reviewer #2: (No Response)

7. PLOS authors have the option to publish the peer review history of their article (what does this mean?). If published, this will include your full peer review and any attached files.

Reviewer #1: No

Reviewer #2: No

---

## [Author Response · Author response to Decision Letter 1]

17 Jan 2022

PONE-D-21-17814R1

Title: Guideline adherence in speech and language therapy in stroke aftercare. A health insurances claims data analysis

Article type: Research Article

**Reviewer #1**

Summary (by reviewer #1)

The authors have carefully responded to the reviewer comments and revised their manuscript accordingly. There are some additional minor concerns that have been identified (below) that could be addressed. Additionally, there are several remaining grammatical issues. It would likely improve the readability of the manuscript if it was proofread/revised in its entirety for English grammar again. If these minor issues are addressed satisfactorily, it is likely that this manuscript could be accepted.

Dear Reviewer, 

Thank you for rereading and commenting on our manuscript. Your detailed hints were again very helpful and useful to revise the paper. The authors discussed the points you mentioned and implemented them as shown below. We have also had the manuscript proofread again by a professional proof reading service.

Best wishes

The authors 

Abstract

In the sentence that starts “Treatment oriented on medical guidelines…”, the word “medical” could be misleading, as it often refers to medication. Perhaps this could be changed to “evidence-based guidelines” or something similar.

Answer (AW): We have changed the wording as you suggested.

Revised sentence: “Treatment oriented on evidence-based guidelines seems likely to improve outcomes.”

As Reviewer 2 points out, using hospitalization duration as a proxy for stroke severity is not ideal. To make the abstract more clearly represent the data, please revise the sentence that starts “Older age and increasing severity of stroke…” by replacing “severity of stroke” with “hospitalization duration”.

AW: We have changed that.

Revised sentence: “Older age and longer hospitalization duration increased the likelihood of guideline recommendations being implemented and earlier initiation of stroke aftercare measures.”

The abstract seems to be missing some of the most salient findings of the paper. Please consider including in the Results section of the Abstract that of the 90.3% of post-stroke patients with an SLT diagnosis, only 44.1% received outpatient SLP services within the first year post-stroke.

AW: Thank you for pointing on that. We included the findings as follows. 

New sentences: “Within the first year after the stroke, 90.3% of patients were diagnosed with speech impairments and swallowing disorders. Overall, 44.1% of patients received outpatient speech and language therapy aftercare.”

Methods

In the Guideline parameters to test for adherence section: From the sentence that starts “Extraction of parameters”, please delete “were involved” from the end of the sentence (Lines 119-121). Also, please add a comma after “medical doctor”.

AW: Thank you. We changed that. 

Revised sentence: “Extraction of parameters to test adherence using claims data was conducted by three scientific researchers: an experienced speech therapist, a medical doctor, and a medical sociologist.”

In the Guideline parameters to test for adherence section: Please add a colon after the sentence that starts “In total, five specific testable…” Otherwise, the sentence that follows is hard to interpret. (Lines 125-126).

AW: We added a colon after the sentence to make clear that a kind of enumeration follows.

Revised sentence: “In total, five specific testable parameters for SLT provision were extracted from the guidelines (Table 1): Patients showing speech and language disorders should be treated by professional speech and language therapists (Parameter 1: speech therapists).”

In the Guideline parameters to test for adherence section: It is still difficult to interpret the sentence that includes, “we defined having at least two specific diagnoses of speech disturbance ...” (Lines 127 – 131). Perhaps you could add the specific ICD-10-GM codes for aphasia and dysarthria (R47.01?) and separate these two conditions in the sentence (e.g., “aphasia (ICD-10-GM code R47.X), dysarthria (ICD-10-GM code R47.X”)) so that the sentence is clearer. Presently, it looks like dysarthria and aphasia have the same ICD-10 code.

AW: We revised that part. For the selection of the data we used the more general three-digit diagnostic code "R47". This includes the diagnostic codes R47.0 (aphasia), R47.1 (dysarthria), and R47.8 (other and unspecified speech and language disorders). We hope that this is now clearer.

New sentence: “To account for spontaneous remission and early inpatient therapy of disorders [22-24], we defined having at least two specific diagnoses of speech disturbance (ICD-10-GM code R47, including aphasia (ICD-10-GM code R47.0), dysarthria (ICD-10-GM code R47.1), and “other and unspecified speech and language disorders” (ICD-10-GM code R47.8)) or swallowing difficulties (dysphagia (ICD-10-GM code R13)) during inpatient or outpatient care as the prerequisite for determining a need of aftercare SLT.”

Results

The last sentence of the Patients with SLT subsection needs to be revised (“Patients receiving aftercare SLT were more frequently and for a longer period in rehabilitation measures after the initial stroke”). (Lines 204-205)

AW: We have divided the sentence into two sentences.

Revised sentences: “In addition, patients who were in inpatient rehabilitation after stroke were more likely to get SLT aftercare. With increasing rehabilitation duration, the likelihood of receiving SLT also increased.”

Discussion

In the second sentence of the first paragraph of the Discussion, it would be helpful to clarify that hospitalization duration was used as a proxy for stroke severity by putting this information in parentheses (i.e., “patients with severe stroke (as captured by hospitalization duration)”). (Lines 256 - 258).

AW: We have added the operationalization of severity in parentheses.

Please make the fourth sentence of the first paragraph of the Discussion section clearer (“Younger patients needed more time for the take-up of aftercare speech therapies.”) (Line 259-260)

AW: We divided the sentence into two sentences and revised them.

Revised sentences: “In the case of younger patients, more time elapsed before the take-up of aftercare speech therapies. In addition, patients of older age were less likely to use SLT aftercare than younger patients.”

Please revise the last sentence of the first paragraph of the Discussion section that starts “A final evaluation…” (Lines 261-262). Perhaps you could start the sentence with, “A deeper look into the data shows that …”, or something similar that hints to the reader what will be discussed in the remainder of the Discussion.

AW: Thank you for your suggestion. We divided the sentence into two sentences.

Revised sentences: “A deeper look into the data shows that the implementation of recommendations is not necessarily related to specific patient groups per se. While some recommendations are implemented well, others for the same group might appear to be in great need of improvement.”

In the first sentence of the Speech therapists subsection of the Discussion, do you mean “multiple diagnoses” rather than “repeated diagnosis”? Could you add swallowing impairments to this sentence?

AW: Thank you. We revised that. 

Revised sentence: “The majority of patients with multiple diagnosis of specific speech, language, and swallowing impairments receives aftercare through speech and language therapists.”

In the first paragraph of the Speech therapists subsection of the Discussion, please revise “Patients with less severe stroke or first stroke” to “Patients with less severe stroke or following an initial stroke”. (Lines 285-286)

AW: We changed that.

Revised sentence: “Patients with less severe stroke or following an initial stroke less frequently received SLT aftercare, or made use of such a therapy, despite a relevant diagnosis.”

In the first paragraph of the Early Initiation and continuity subsection of the Discussion, please change “therapy resumption” in the first sentence to “therapy initiation.” (Line 324)

AW: Thank you. We changed that. 

New sentence: “One aspect of guideline-adherent therapy after stroke is early therapy initiation for stroke patients after hospital discharge.”

Tables

On Table 2, it might be helpful to revise the 7th row description from “Speech, language disorders: R47, R13” to “Dysarthria, aphasia, or dysarthria” (and add the relevant ICD-10 codes). Please include the raw number and percentage of the population for each of the three codes in the table (e.g., How many individuals had dysphagia?).

AW: Thank you for your suggestion. We have adjusted the labeling in the table and added the footnote to row 7 regarding the diagnosis codes. Unfortunately, a breakdown for the individual diagnoses is no longer possible. We really would have liked to have the individual disorder diagnoses reported separately. But, the data were aggregated in the course of data preparation and various diagnoses were combined. The aggregation of the data was a requirement of the data donors (health insurance companies) because this makes it more difficult to re-identify individual patients. We regret not being able to implement this advice.

7. PLOS authors have the option to publish the peer review history of their article (what does this mean?). If published, this will include your full peer review and any attached files.

Do you want your identity to be public for this peer review? For information about this choice, including consent withdrawal, please see our Privacy Policy.

Reviewer #1: No

Reviewer #2: No

---

## [Editor Report · Decision Letter 2]

19 Jan 2022

Guideline adherence in speech and language therapy in stroke aftercare. A health insurance claims data analysis.

PONE-D-21-17814R2

Dear Dr. Schindel,

We’re pleased to inform you that your manuscript has been judged scientifically suitable for publication and will be formally accepted for publication once it meets all outstanding technical requirements.

Kind regards,

Marie Jetté

Academic Editor

PLOS ONE
---

## [Editor Report · Acceptance letter]

25 Jan 2022

PONE-D-21-17814R2 

Guideline adherence in speech and language therapy in stroke aftercare. A health insurance claims data analysis. 

Dear Dr. Schindel:

I'm pleased to inform you that your manuscript has been deemed suitable for publication in PLOS ONE. Congratulations! Your manuscript is now with our production department. 

Kind regards, 

on behalf of

Dr. Marie Jetté 

Academic Editor

PLOS ONE